# Coverage-centric Coreset Selection for High Pruning Rates

**Haizhong Zheng, Rui Liu, Fan Lai, Atul Prakash**
Computer Science and Engineering
University of Michigan
Ann Arbor, MI 48109, USA
{hzzheng,ruixliu,fanlai,aprakash}@umich.edu

## Abstract

One-shot coreset selection aims to select a representative subset of the training data, given a pruning rate, that can later be used to train future models while retaining high accuracy. State-of-the-art coreset selection methods pick the highest importance examples based on an importance metric and are found to perform well at low pruning rates. However, at high pruning rates, they suffer from a *catastrophic accuracy drop*, performing worse than even random sampling. This paper explores the reasons behind this accuracy drop both theoretically and empirically. We first propose a novel metric to measure the coverage of a dataset on a specific distribution by extending the classical geometric set cover problem to a distribution cover problem. This metric helps explain why coresets selected by SOTA methods at high pruning rates perform poorly compared to random sampling because of worse data coverage. We then propose a novel one-shot coreset selection method, *Coverage-centric Coreset Selection (CCS)*, that jointly considers overall data coverage upon a distribution as well as the importance of each example. We evaluate CCS on five datasets and show that, at high pruning rates (e.g., 90%), it achieves significantly better accuracy than previous SOTA methods (e.g., at least 19.56% higher on CIFAR10) as well as random selection (e.g., 7.04% higher on CIFAR10) and comparable accuracy at low pruning rates. We make our code publicly available at GitHub[1].

## 1 Introduction

One-shot coreset selection aims to select a small subset of the training data that can later be used to train future models while retaining high accuracy (Coleman et al., 2019; Toneva et al., 2018). One-shot coreset selection is important because full datasets can be massive in many applications and training on them can be computationally expensive. A favored way to select coresets is to assign an importance score to each example and select more important examples to form the coreset (Paul et al., 2021; Sorscher et al., 2022). Unfortunately, current SOTA methods for one-shot coreset selection suffer *a catastrophic accuracy drop* under high pruning rates (Guo et al., 2022; Paul et al., 2021). For example, for the CIFAR-10, a SOTA method (forgetting score (Toneva et al., 2018)) achieves 95.36% accuracy with a 30% pruning rate, but that accuracy drops to only 34.03% at a 90% pruning rate (which is significantly worse than random coreset selection). This accuracy drop is currently unexplained and limits the extent to which coresets can be practically reduced in size.

In this paper, we provide both theoretical and empirical insights into reasons for the catastrophic accuracy drop and propose a novel coreset selection algorithm that overcomes this issue. We first extend the classical geometrical set cover problem to a density-based distribution cover problem and provide theoretical bounds on model loss as a function of properties of a coreset providing specific coverage on a distribution. Furthermore, based on theoretical analysis, we propose a novel metric $AUC_{pr}$, which allows us to quantify how a dataset covers a specific distribution (Section 3.1). With the proposed metric, we show that coresets selected by SOTA methods at high pruning rates have much worse data coverage than random pruning, suggesting a linkage between poor data coverage

---

[1] https://github.com/haizhongzheng/Coverage-centric-coreset-selection

of SOTA methods and poor accuracy at high pruning rates. We note that data coverage has also been studied in active learning setting (Ash et al., 2019; Citovsky et al., 2021), but techniques from active learning do not trivially extend to one-shot coreset selection. We discuss the similarity and differences in Section 5.

We then propose a novel algorithm, *Coverage-centric Coreset Selection (CCS)*, that addresses catastrophic accuracy drop by improving data coverage. Different from SOTA methods that prune unimportant (easy) examples first, CCS is inspired by stratified sampling and guarantees the sampling budget across importance scores to achieve better coverage at high pruning rates. (Section 3.3).

We find that CCS overcomes catastrophic accuracy drop at high pruning rates, outperforming SOTA methods by a significant margin, based on the evaluation on five datasets (CIFAR10, CIFAR100 (Krizhevsky et al., 2009), SVHN (Netzer et al., 2011), CINIC10 (Darlow et al., 2018), and ImageNet (Deng et al., 2009)). For example, at 90% pruning rate, on the CIFAR10 dataset, CCS achieves 85.7% accuracy versus 34.03% for a SOTA coreset selection method based on forgetting scores (Toneva et al., 2018). Furthermore, CCS also outperforms random selection at high pruning rates. For example, at a 90% pruning rate, CCS achieves 7.04% and 5.02% better accuracy than random sampling on CIFAR10 and ImageNet, respectively (Section 4). Our method also outperforms a concurrent work called Moderate (Xia et al., 2023) on CIFAR10 by 5.04% and ImageNet by 5.20% at 90% pruning rate. At low pruning rates, CCS still achieves comparable performance to baselines, outperforming random selection.

To summarize, our contributions are as follows:

1. We extend the geometric set cover problem to a density-based distribution cover problem and provide a theoretical bound on model loss as a function of properties of a coreset (Section 3.1, Theorem 1).

2. We propose a novel metric $\text{AUC}_{pr}$ to quantify data coverage for a coreset (Section 3.1). As far as we know, $\text{AUC}_{pr}$ is the *first* metric to measure how a dataset covers a distribution.

3. Based on this metric, we show that SOTA coreset selections tend to have poor data coverage at high pruning rates – worse than random selection, thus suggesting a linkage between coverage and observed catastrophic accuracy drop (Section 3.2).

4. To improve coverage in coreset selection, we propose a novel one-shot coreset selection method, CCS, that uses a variation of stratified sampling across importance scores to improve coverage (Section 3.3).

5. We evaluate CCS on five different datasets and compare it with six baselines, and we find that CCS significantly outperforms baselines as well as random coreset selection at high pruning rates and has comparable performance at low pruning rates (Section 4).

Based on our results, we consider CCS to provide a new strong baseline for future one-shot coreset selection methods, even at higher pruning rates.

## 2 PRELIMINARIES

### 2.1 ONE-SHOT CORESET SELECTION

Consider a classification task with a training dataset containing $N$ examples drawn i.i.d. from an underlying distribution $P$. We denote the training dataset as $\mathcal{S} = \{(\mathbf{x}_i, y_i)\}_{i=1}^N$, where $\mathbf{x}_i$ is the data, and $y_i$ is the ground-truth label. The goal of one-shot coreset selection is to select a training subset $\mathcal{S}'$ given a pruning rate $\alpha$ *before training* to maximize the accuracy of models trained on this subset, which can be formulated as the following optimization problem (Sener & Savarese, 2017):

$$\min_{\mathcal{S}' \subset \mathcal{S}: \frac{|\mathcal{S}'|}{|\mathcal{S}|} \leq 1-\alpha} \mathbb{E}_{\boldsymbol{x}, y \sim P}[l(\boldsymbol{x}, y; h_{\mathcal{S}'})], \tag{1}$$

where $l$ is the loss function, and $h_{\mathcal{S}'}$ is the model trained with a labelled dataset $\mathcal{S}'$.

SOTA methods typically assign an importance score (also called a difficulty score or importance metric) to each example and preferably select more important (difficult) examples to form the coreset. One proposed importance score is the Forgetting score (Toneva et al., 2018), which is defined as

the number of times an example is incorrectly classified after having been correctly classified earlier during model training. Another is area under the margin (AUM) (Pleiss et al., 2020), which is a data difficulty metric that identifies mislabeled data. Another importance metric, EL2N score (Paul et al., 2021) estimates data difficulty by the L2 norm of error vectors.

SOTA algorithms based on these metrics all prune examples with low importance first (Sorscher et al., 2022). As discussed in Sorscher et al. (2022), harder examples usually contain more information and are treated as more important data and thus more desirable for inclusion in a coreset.

## 2.2 CATASTROPHIC ACCURACY DROP WITH HIGH PRUNING RATES

Although SOTA coreset methods (Toneva et al., 2018; Coleman et al., 2019; Paul et al., 2021) report encouraging performance at low pruning rates, a *catastrophic accuracy drop* is observed as pruning rate increases (Guo et al., 2022; Paul et al., 2021). For instance, as shown in Figure 1, SOTA methods based on forgetting number and AUM outperform random sampling when the pruning rate is less than 70% but considerably underperform random sampling at 90% pruning rate, achieving only 34.03% and 28.06% accuracy, respectively, which are much worse than 79.04% for random sampling.

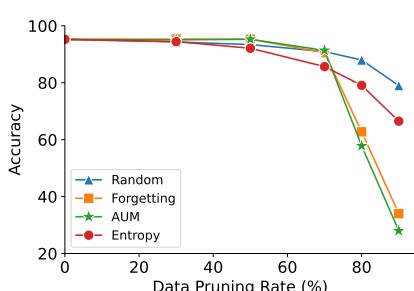

Figure 1: Existing coreset solutions have better accuracy than random sampling at low pruning rates, but perform worse at high pruning rates.

We hypothesize that this accuracy drop is a result of bad data coverage caused by the biased pruning of importance-based methods. In this paper, we conduct an in-depth study on this catastrophic accuracy drop and propose a novel coreset selection method that overcomes the catastrophic accuracy drop at high pruning rates.

## 3 THE COVERAGE OF CORESETS

We start by studying data coverage provided by different coresets. We first extend the classical geometric set cover to a density-based distribution cover problem and propose a novel metric to compare the data coverage of different coresets (Section 3.1). We then show that biased pruning on easy data leads to poor data coverage of coresets with high pruning rates (Section 3.2). In Section 3.3, inspired by stratified sampling, we propose a novel coverage-centric coreset selection method which jointly consider both coverage and importance of each example based on a given importance metrics.

### 3.1 DENSITY-BASED PARTIAL COVERAGE

To compare the data coverage of different coreset methods, we need to quantitatively measure how well a dataset $\mathcal{S}$ covers a distribution $P$. In classical geometric set cover setting (Fowler et al., 1981; Sener & Savarese, 2017), we say a set $\mathcal{S}'$ is a $r$-cover of another set $\mathcal{S}$, when a set of $r$-radius balls centered at each element in $\mathcal{S}'$ covers the entire $\mathcal{S}$. The radius $r$ can be used as a metric to measure coverage of $\mathcal{S}'$ on $\mathcal{S}$. However, how to measure the coverage of a dataset on a distribution is not well-studied.

To measure how well a set covers a distribution, we extend the classical geometric set cover to the *density-based distribution cover*. Instead of covering a set, we study the covering on a distribution $P_\mu$ and consider probability density in different areas of the input space. Instead of taking a complete cover, we introduce the cover percentage $p$ to describe how a set covers different percentages of a distribution to better understand the trade-off between cover radius $r$ and cover percentage $p$:

**Definition 1** ($p$-partial $r$-cover). *Given a metric space $(X, d)$, a set $\mathcal{S} \subset X$ is a p-partial r-cover for a distribution $P_\mu$ on the space $X$ if:*

$$\int_X \mathbf{1}_{\cup_{\mathbf{x} \in \mathcal{S}} B_d(\mathbf{x}, \mathbf{r})}(\boldsymbol{x}) d\mu(\boldsymbol{x}) = p,$$

*where $B_d(\boldsymbol{x}, r) = \{\boldsymbol{x}' \in X : d(x, x') \leq r\}$ is a r-radius ball whose center is $\boldsymbol{x}$, p measures coverage (which we will usually express as a percentage and refer to as* percentage coverage*), and*

$\mu$ *is the probability measure*[2] *corresponding to probability density function* $P_\mu$. *The probability measure* $\mu$ *represents the density information in the input space.*

For a model trained on a $r$-cover coreset, Sener & Savarese (2017) proved a bounded risk on the entire training dataset. In Theorem 1 below, we extend their result to the distribution coverage scenario. The proof of Theorem 1 relies on the same assumptions as in Sener & Savarese (2017), in particular Lipschitz-continuity loss function and zero training error on the coreset. The zero training error assumption may not always be realistic, but Sener & Savarese (2017) found that the resulting bounds still hold and we rely on the same assumption. The proof of theorem 1 can be found in the Appendix C.

**Theorem 1.** *Given n i.i.d. samples drawn from* $P_\mu$ *as* $\mathcal{S} = \{\boldsymbol{x}_i, y_i\}_{i \in [n]}$ *where* $y_i \in [C]$ *is the class label for example* $x_i$, *a coreset* $\mathcal{S}'$ *which is a p-partial r-cover for* $P_\mu$ *on the input space* $X$, *and an* $\epsilon > 1 - p$, *if the loss function* $l(\cdot, y, \boldsymbol{w})$ *is* $\lambda_l$*-Lipschitz continuous for all* $y$, $\boldsymbol{w}$ *and bounded by L, the class-specific regression function* $\eta_c(\boldsymbol{x}) = p(y = c|\boldsymbol{x})$ *is* $\lambda_\eta$*-Lipschitz for all c, and* $l(\boldsymbol{x}, y; h_{\mathcal{S}'}) = 0$, $\forall (\boldsymbol{x}, y) \in \mathcal{S}'$, *then with probability at least* $1 - \epsilon$:

$$\left| \frac{1}{n} \sum_{\boldsymbol{x}, y \in \mathcal{S}} l(\boldsymbol{x}, y; h_{\mathcal{S}'}) \right| \leq r(\lambda_l + \lambda_\eta LC) + L\sqrt{\frac{\log \frac{p}{p+\epsilon-1}}{2n}}. \quad (2)$$

Theorem 1 shows that the model trained on a $p$-partial $r$-cover coreset $\mathcal{S}'$ has a bounded risk on the entire training dataset $\mathcal{S}$. There are several implications of Theorem 1: 1) A model trained on a $p$-partial $r$-cover coreset has a training risk bounded with the covering radius $r$ and an additional term. This term goes to zero as $n$ increases to infinity. Also, we want a small $\epsilon$, a large $p$ and a small radius $r$ to get a high probability small bound on the loss. 2) The probability that Equation 2 holds is bounded by cover percentage $p$ (since $1 - \epsilon < p$). This indicates that the bound in Theorem 1 becomes meaningless (low probability) when the cover percentage $p$ is too small. 3) Theorem 1 is a more general form of Theorem 1 in Sener & Savarese (2017); if we set $p = 1$ (i.e., complete cover), these two theorems are the same.

Definition 1 shows that, for a given set $\mathcal{S}$ and a distribution $P_\mu$, the cover percentage $p$ increases with the cover radius $r$. Conversely, given a cover percentage $p$ and a set $S$, we can get a minimum cover radius $r$ for the $\mathcal{S}$ to achieve $p$ percent coverage on distribution $P_\mu$. For a given set $\mathcal{S}$ and a distribution $P_\mu$, we can plot a $p$-$r$ curve to represent this relationship. In practice, the distribution $P_\mu$ is unknown. So, we estimate the cover percentage $p$ for a given subset $S$ of the training dataset and a radius $r$ by measuring the fraction of the test dataset that is covered [3]. To assess coverage for a radius $r$, we use the L2 distance between activations of the final convolutional layer of a model trained with the entire dataset as the distance metric $d$, as suggested in Babenko & Lempitsky (2015); Sharif Razavian et al. (2014); Hsieh et al. (2018).

Figure 2 shows the $p$-$r$ curves for several subsets of CIFAR10 training dataset: (1) entire dataset (no pruning); (2) random coreset at 90% pruning rate (i.e., random 10% of the training data); and (3) coreset selected based on forgetting score as the importance metric, as proposed by Toneva et al. (2018). As expected, the required radius $r$

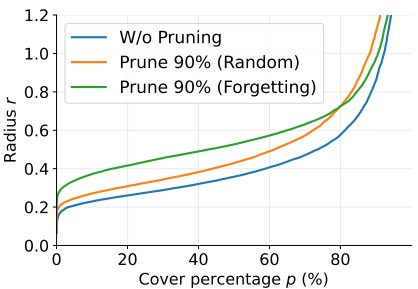

Figure 2: The $p$-$r$ curves of different subsets of CIFAR10 training dataset. The dataset without any pruning has the lowest curve (blue). Forgetting curve (green) and random curve (orange) have a crossover around $p = 80\%$.

increases to get the desired cover percentage $p$ for all coresets. Also coreset with no pruning has the lowest curve, as expected. In general, lower curves are better and we expect models based on

---

[2]By Radon–Nikodym theorem, there is a probability measure $\mu$: $P_\mu(x)dx = d\mu(x)$.

[3]We noticed that some misclassified examples have a much larger minimum cover radius than other data. To eliminate the influence of outliers, when we plot $p$-$r$ curves, we ignore the data misclassified by the model trained with the entire CIFAR10 dataset (with a 95.25% accuracy).

the corresponding datasets to have lower loss. However, we find that there is crossover between forgetting curve and random curve around $p = 80\%$, making a comparison less obvious.

To make a more obviously quantitative comparison, we propose to use area under the $p$-$r$ curve, $AUC_{pr}$, as a proxy metric to assess the quality of a coreset selection strategy. Lower values of $AUC_{pr}$ suggest better coverage by the coreset. We note that $AUC_{pr}$ is the expected minimum distance between examples following the underlying distribution $P_\mu$ to those in the coreset, as stated in the following proposition (proved in Appendix C):

**Proposition 1.** *Given a metric space* $(X, d)$, *a distribution* $P_\mu$, *cover percentage* $p$, *a set* $S$, *and* $f_r(p) : [0, 1] \rightarrow [0, +\infty]$ *representing the mapping between* $p$ *and* $r$, *if* $f_r$ *is Riemann-integrable, the AUC of the* $p$-$r$ *curve* $AUC_{pr}(S) = \int_0^1 f_r(p) dp = \mathbb{E}_{x \sim P_\mu}[d(S, \boldsymbol{x})]$, *where* $d(S, \boldsymbol{x}) = \min_{\boldsymbol{x}' \in S} d(\boldsymbol{x}', \boldsymbol{x})$.

In practice, we can assess $AUC_{pr}$ with test set $D_{test}$ : $AUC_{pr}(S) = \mathbb{E}_{\boldsymbol{x} \in D_{test}}[\min_{\boldsymbol{x}' \in S} d(\boldsymbol{x}', \boldsymbol{x})]$. We also note that this expected minimum distance (and thus AUC) is related to $r$ in Theorem 1; lower values of $AUC_{pr}$ are preferred for lower loss and thus higher model accuracy.

## 3.2 COVERAGE ANALYSIS ON CORESETS

Table 1: $AUC_{pr}$ of different methods with different pruning rates. With low pruning rates (30%, 50%), forgetting and AUM have similar $AUC_{pr}$ to random. With high pruning rates (80%, 90%), forgetting and AUM have larger $AUC_{pr}$ than random, which means worse data coverage.

| Pruning Rate | 30% | 50% | 70% | 80% | 90% |
|---|---|---|---|---|---|
| Random | 0.496 | 0.509 | 0.532 | 0.552 | 0.589 |
| Forgetting | 0.492 | 0.511 | 0.551 | 0.580 | 0.631 |
| AUM | 0.496 | 0.518 | 0.558 | 0.586 | 0.646 |

Table 1 shows the $AUC_{pr}$ for different CIFAR10 coresets with different pruning rates. With a 30% or 50% pruning rate, forgetting and AUM have similar $AUC_{pr}$ to random sampling. However, with a pruning rate larger than 70%, forgetting and AUM have larger $AUC_{pr}$ than random pruning, suggesting that forgetting and AUM can be expected to have a worse accuracy than random pruning.

The reason behind higher $AUC_{pr}$ for forgetting and AUM is that they prune easy examples, which tend to be located in the high-density area of distribution since more common examples often tend to be well-separated easier examples for classifiers. Low pruning rate coresets still contain plenty of easy examples for all methods; thus coverage is not a significant factor at low pruning rates. However, with high pruning rates, coresets can completely exclude examples from the high-density areas, reducing coverage and raising $AUC_{pr}$, and thus potentially raising loss and reducing accuracy.

For the data points in Table 1, Figure 3 plots model accuracy and $AUC_{pr}$. Size of the ball at a point represents pruning rate for that point – larger ball size indicates a larger pruning rate. The plot generally confirms our intuitions from Section 3.1 that higher $AUC_{pr}$ values tend to lead to lower model accuracy.

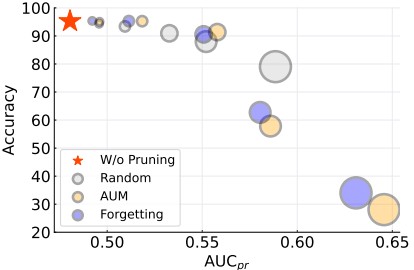

Figure 3: The relationship between $AUC_{pr}$ and accuracy. Different colors stands for different coreset selected methods. A larger circle size indicates a higher pruning ratio on the dataset. Smaller $AUC_{pr}$ often ends up with better accuracy. With high pruning rates (larger balls), forgetting and AUM have a higher $AUC_{pr}$ and worse accuracy.

## 3.3 METHODOLOGY: COVERAGE-CENTRIC CORESET SELECTION

Based on our density-based distribution setting in Section 3.1, if the sampling budget is limited, easy examples in the high-density area provide more coverage than hard examples in the low-density area.

---

**Algorithm 1:** Coverage-centric Coreset Selection (CCS)

---

1 **Inputs:**
   $\mathbb{S} = \{(x_i, y_i, s_i)\}_{i=1}^n$: dataset with the importance score for each example;
   $\alpha$: dataset pruning rate; $\beta$: hard cutoff rate ($\beta \leq 1 - \alpha$); $k$: the number of strata.

---

2 $\mathbb{S}' \leftarrow \mathbb{S} \setminus \{\lfloor n * \beta \rfloor \text{ hardest examples}\}$ ;         /* Prune hard examples first */
3 $R_1, R_2, ..., R_k \leftarrow$ Split scores in $\mathbb{S}'$ into $k$ ranges with an even range width;
4 $\mathcal{B} \leftarrow \{\mathbb{B}_i, : \mathbb{B}_i \text{ consists of examples whose scores are in } R_i, i = 1...k\}$;
5 $m \leftarrow n \times (1 - \alpha)$ ;         /* m is total budget across all strata */
6 $\mathbb{S}_c \leftarrow \varnothing$ ;                              /* Initialize the coreset */
7 **while** $\mathcal{B} \neq \varnothing$ **do**
8 $\quad$ $\mathbb{B}_{min} \leftarrow \underset{\mathbb{B} \in \mathcal{B}}{\arg \min} |\mathbb{B}|$ ;  /* Select the stratum with fewest examples */
9 $\quad$ $m_B \leftarrow \min\{|\mathbb{B}_{min}|, \lfloor \frac{m}{|\mathcal{B}|} \rfloor\}$ ; /* Compute budget for selected stratum */
10 $\quad$ $\mathbb{S}_B \leftarrow$ randomly sample $m_B$ examples from $\mathbb{B}_{min}$ ;
11 $\quad$ $\mathbb{S}_c \leftarrow \mathbb{S}_c \cup \mathbb{S}_B$ ;                              /* Update the coreset */
12 $\quad$ $\mathcal{B} \leftarrow \mathcal{B} \setminus \{\mathbb{B}_{min}\}$ ;                  /* Done with selected straum */
13 $\quad$ $m \leftarrow m - m_B$ ;       /* Update total budget for remaining strata */
14 **return** $\mathbb{S}_c$ ;                              /* Return the final coreset */

---

SOTA methods prune low-importance (easy) data from the high-density area, which contributes to a larger $\text{AUC}_{pr}$ value and thus possibly lower model accuracy at high pruning rates.

Inspired by this insight, we propose a novel coreset selection method called Coverage-centric Coreset Selection (CCS), presented in Algorithm 1. CCS is still based on existing importance scores. However, unlike SOTA methods that simply select important examples first, CCS aims to still guarantee the data coverage on high-density area at high pruning rates to improve coverage and lower the value of $\text{AUC}_{pr}$.[4]

Compared to SOTA methods, CCS still assigns the sampling budget to the high-density area containing easy examples at high-pruning rates, which provides larger coverage to the underlying distribution. Compared to random sampling, CCS assigns a larger sampling budget to the low-density area, where hard examples are informative for training.

CCS first divides the dataset into different non-overlapping strata based on importance scores. Each stratum has a fixed-length score range, but may include different numbers of examples. We fix an initial budget on the number of examples to be chosen from each strata, based on the desired pruning rate, but, if a stratum has fewer examples than the budget, remaining budget is evenly assigned to other strata.

Thus, stratified sampling differs from random sampling in that examples from "dense" stratum are less likely to be sampled than examples in a "sparse" stratum. Also, with a low pruning rate, stratified sampling tends to first discard data from low-importance strata, similar to SOTA methods. At high pruning rates, stratified sampling diverges from SOTA methods in that low-importance strata still get budget to guarantee data coverage. (In Appendix D.5, we discuss another coreset selection method based on importance sampling.)

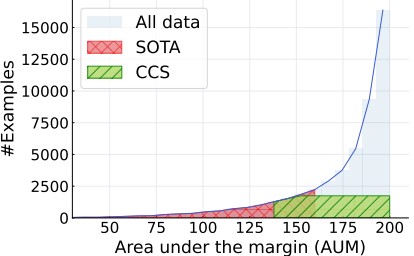

Figure 4: CIFAR10 data distribution with AUM as the importance metric (lower AUM values are more important). At 70% pruning rate, SOTA method selects data from the red region, since it prune low-importance (easy) examples first. Our method, CCS, selects data from the green region using stratified sampling across the importance metric, along with pruning hardest examples on the left. The coreset selected by the SOTA method lacks "easy" examples on the right in the high-density region.

---

[4]Further evaluations confirm that coresets selected by CCS have a lower $\text{AUC}_{pr}$ (i.e., better coverage) than the SOTA method, which prunes easy examples first. See Appendix D.2.

Before CCS assigns budgets to strata, CCS prunes $\beta$ percent "hard" examples based on importance scores (Line 2 in Algorithm 1). This is based on two insights: (1) Mislabeled examples often also have higher importance scores (Swayamdipta et al., 2020) but do not benefit accuracy. As the pruning rate increases, the percentage of mislabeled data in the coreset also rises, which hurts model accuracy. (2) When data is really scarce, we need more data from higher-density areas to get better coverage. Pruning hard examples helps allocate more budget to high-density strata. In practice, we use grid search to find $\beta$ as a hyperparameter. We discuss how $\beta$ impacts training in Section 4.2 and further details on selecting $\beta$ in Appendix B.

In Figure 4, we show an example on CIFAR10 to compare the fraction of data selected between a SOTA method (AUM) and CCS (also based on AUM) for 70% pruning rate. The data selected by the SOTA method (Pleiss et al., 2020), shown with red shading, prunes all the easy examples (high AUM value). In contrast, CCS selects the data in the green region. Notice that CCS selects slightly fewer examples in the curved area (around 140) because those stratum had insufficient examples for the budget; excess budget is evenly assigned to other strata.

## 4 EVALUATION

We evaluate CCS against six baselines (random, entropy, forgetting, EL2N, AUM, and Moderate) on five datasets (CIFAR10, CIFAR100, SVHN, CINIC10, and ImageNet). In total, we trained over 800 models for performance comparison. We find that CCS outperforms other baselines at high pruning rates while achieving comparable accuracy at low pruning rates. We conduct an ablation study (Section 4.2) to better understand how hyperparameters and different components influence the accuracy. Due to space limits, we include experimental setting in Appendix B. All training is repeated 5 times with different random seeds to calculate mean accuracy with standard deviation.

**Baselines.** We compare CCS with six baselines, with the latter five being SOTA methods: 1) **Random**: Select examples with uniform random sampling to form a coreset. 2) **Entropy** (Coleman et al., 2019): Select highest entropy examples for coreset, since entropy reflects uncertainty of training examples and thus more importance to training. 3) **Forgetting score** (Toneva et al., 2018): Select examples with highest Forgetting scores, which is the number of times an example is incorrectly classified after being correctly classified earlier during model training; 4) **EL2N** (Paul et al., 2021): Select examples with highest EL2N scores, which estimate data difficulty or importance by the L2 norm of error vectors. 5) **Area under the margin (AUM)** (Pleiss et al., 2020): Select examples with highest AUM scores, which measures the probability gap between the target class and the next largest class across all training epochs. A larger AUM suggests higher difficulty and importance. 6) **Moderate** (Xia et al., 2023) is a concurrent work that proposes a distance-based score for one-shot coreset selection. Moderate treats examples close to the median value of feature space as more important examples. For all baselines, we prune unimportant examples first when selecting coresets, based on a chosen importance metric. In Section 4.1, we combine CCS with AUM score, and we also discuss the performance of combining CCS with other importance metrics in Appendix D.

### 4.1 CORESET PERFORMANCE COMPARISON

As shown in Figure 5, our proposed method CCS significantly outperforms SOTA schemes as well as the random selection at high pruning rates by comparing CSS with six other baselines on three datasets (CIFAR10, CIFAR100, and ImageNet). We observe that, besides Moderate, other four SOTA coreset methods tend to have similar or higher accuracy than random at 50% or lower pruning rates, but they have worse performance than random at high pruning rates (e.g., 80% and 90%). CCS achieves better accuracy than all other baseline selection methods at high pruning rates and achieves a comparable performance at low pruning rates. For example, at a 90% pruning rate of CIFAR10, CCS achieves 7.04% better accuracy than random sampling. And CCS achieves 5.08% better accuracy than random sampling on ImageNet at a 90% pruning rate. We have similar findings on SVHN and CINIC10 (detailed results in the Appendix D). CCS outperforms all baselines at high pruning rates on all these datasets.

### 4.2 ABLATION STUDY & ANALYSIS

We first observe that, when $k = 1$ and $\beta = 0$, CCS is equivalent to random sampling, i.e., random sampling is a special case of CCS. This guarantees that CCS, with appropriate choice of hyperpa-

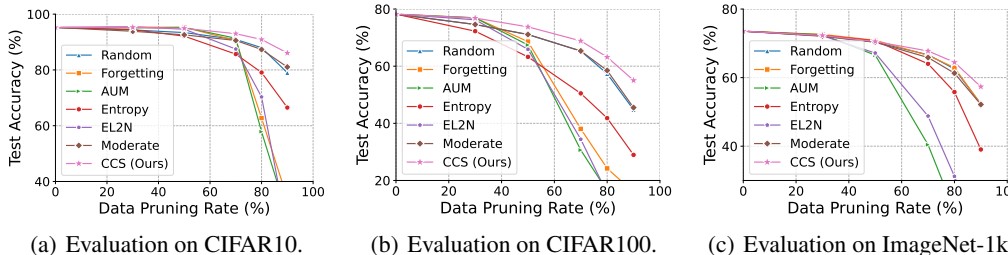

(a) Evaluation on CIFAR10.  (b) Evaluation on CIFAR100.  (c) Evaluation on ImageNet-1k.

Figure 5: Performance comparison between our proposed method (CCS) and other baselines on CIFAR10, CIFAR100, and ImageNet-1k. The pruning rate is the fraction of examples removed from the original datasets. The evaluation results show that our method achieves better performance than all other baselines at high pruning rates (e.g., 70%, 80%, 90%) and comparable performance at low pruning rates (e.g., 30%, 50%). We also present detailed numerical numbers in Appendix D.

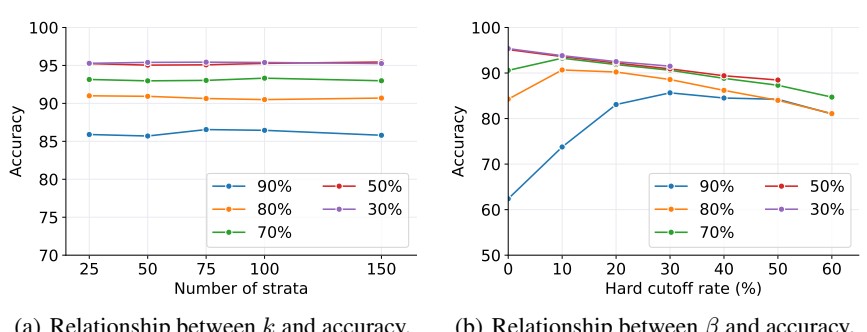

(a) Relationship between $k$ and accuracy.  (b) Relationship between $\beta$ and accuracy.

Figure 6: Ablation study on the number of strata ($k$) and hard cutoff rate ($\beta$). We apply CCS with the forgetting score on CIFAR10. Each curve is corresponding to a pruning rate. Different $k$ leads to similar accuracy. The optimal $\beta$ goes up as the pruning rate increases.

rameters, will not underperform random sampling. In practice, larger values of $k$ tend to be more useful to get the stratified sampling effect and to outperform random. Also, as shown in Figure 6(a), for large k values, we found that the number of strata is not a sensitive hyperparameter.

We then study the hard cutoff rate $\beta$. From Figure 6(b), we observe that the optimal hard cutoff rate $\beta$ goes up as the pruning rate increases. This is expected since a larger hard cutoff rate helps shift more budget to high-density strata (see Section 3.3), When the total data budget is limited, better coverage on high-density strata helps improve model performance. We also observe that the optimal $\beta$ is 0 when the pruning rates is 30% and 50%. A possible reason is that the percentage of mislabeled data is low in CIFAR10 (Pleiss et al., 2020), and deep neural network is robust to a small portion of noisy data (Rolnick et al., 2017). Grid search for $\beta$ can introduce extra costs in selecting effective coresets. However, we note that our paper shows the possibility of achieving better accuracy than random sampling at a high pruning rate. We leave how to efficiently finetune $\beta$ for future work.

Table 2: Ablation study on different components of coverage-centric coreset select method for importance metric of forgetting score. Best accuracy is in bold in each column.

| Pruning Rate | 30% | 50% | 70% | 80% | 90% |
|---|---|---|---|---|---|
| Random | 94.33 | 93.4 | 90.94 | 87.98 | 79.04 |
| Forgetting | 95.36 | **95.29** | 90.56 | 62.74 | 34.03 |
| Stratified only | 95.23 | 95.13 | 90.78 | 81.82 | 59.35 |
| Prune hard only | 91.50 | 88.46 | 85.62 | 80.98 | 80.62 |
| CCS (Stratified + Prune hard) | **95.40** | 95.04 | **92.97** | **90.93** | **85.7** |

Next, we study the impact of stratified sampling and pruning hard examples on model accuracy. We measure the forgetting score for CIFAR10 and then apply only stratified sampling or only pruning hard examples to select the coreset, respectively, in Table 2. By comparing stratified sampling only with the vanilla forgetting method (pruning easy first), we observe that applying stratified sampling improves accuracy compared to the vanilla forgetting method, but it is still inferior to random sampling when we prune $80\%$, or $90\%$ fraction of examples. Sorscher et al. (2022) discussed that pruning hard data is a better strategy than pruning easy data when the original entire data is scarce. In our evaluation, we observe that pruning hard data outperforms pruning easy data at high pruning rates, but still has worse accuracy than CCS.

We also considered using a proportional importance sampling strategy to improve coverage that samples examples *proportional* after mapping importance score to a probability for one dataset. Unfortunately, the strategy underperformed random sampling at high pruning rates (results in Appendix D.5). This evaluation suggested it is non-trivial to beat random at high pruning rates, even for a strategy that improves coverage.

## 5 RELATED WORK

Besides the coreset methods based on entropy (Coleman et al., 2019), forgetting score (Toneva et al., 2018), AUM (Pleiss et al., 2020), and EL2N scores (Paul et al., 2021) discussed in Section 2.1, Sorscher et al. (2022) use $k$-means to estimate importance scores, and Sener & Savarese (2017) apply greedy $k$-center to choose the coreset with good data coverage. However, $k$-means and $k$-center algorithms need to calculate the distance matrix between examples, which leads to an $O(n^2)$ time complexity and are expensive to scale to large datasets.

Data selection is also studied in active learning and some methods recognize importance of diversity or coverage in data selection (Ash et al., 2019; Citovsky et al., 2021; Beluch et al., 2018; Hsu & Lin, 2015; Chang et al., 2017). While our scheme shares the idea of improving coverage with these methods, one-shot coreset selection philosophically differs from dataset selection in active learning: one-shot coreset selection aims to select a *model-agnostic coreset* for training new models from scratch, while active learning aims to select a subset that, given the current state of the model, is to improve the model in the next training round. For example, BADGE (Ash et al., 2019), a diversity-based selection method in active learning, selects a subset by performing k-means++ on the gradient embedding based on the latest model and then repeats that for subsequent rounds during training.

Another significant difference is that in active learning, total number of examples queried across all rounds of training can be much larger than the size of a one-shot coreset. For instance, in experimental results of Ash et al. (2019), BADGE queried a different set of 10K (20%) examples on every round for the CIFAR-10 dataset. The total number of different examples queried across all rounds is even larger. In contrast, CCS finds a fixed small subset (e.g. 10% data) that can be used to train new models from scratch while achieving high accuracy.

Another popular coreset selection area is the coreset selection for classical clustering problems such as k-means and k-median (Feldman et al., 2020; Feldman & Langberg, 2011; Cohen-Addad et al., 2021; Huang et al., 2019). However, these coreset methods are usually designed based on the good mathematical property of classical clustering. For example, importance sampling is a common practice for coreset for clustering (Huang & Vishnoi, 2020), but it does not generalize well in the deep learning classification scenario.

## 6 CONCLUSION

We conduct an in-depth study to understand the *catastrophic accuracy drop* issue of one-shot coreset selection. By extending classical geometric set cover to density-based partial cover, we propose a novel data coverage metric and show that existing importance-based coreset selections lead to poor data coverage, which in turn degrades model performance. Based on this observation, we design a novel Coverage-centric Coreset Selection (CCS) method to improve the data coverage, which empirically outperforms other baseline methods on five datasets. Our paper shows that it is possible to achieve much better accuracy than random sampling, even in a high pruning rate setting, and provides a strong baseline for future research.

REPRODUCIBILITY STATEMENT

We also include detailed experimental setting in the appendix, and the code is available at GitHub.

ACKNOWLEDGEMENTS

We thank Yuwei Bao, Tony Zhang, and Yiwen Zhang for helpful discussions. We thank the anonymous reviewers and area chairs for their time and efforts during the reviewing process. Their feedback help us better improve our paper. This work is partially supported by DARPA under agreement number 885000. Any opinions, findings, and conclusions or recommendations expressed in this material are those of the author(s) and do not necessarily reflect the views of our research sponsors.

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

## A    OVERVIEW

This appendix provides details on our experimental setting, theoretical analysis, and additional evaluation results. In Section B, we introduce the detailed setup of our experiment to improve the reproducibility of our paper. In Section C, we provide proof for Theorem 1 and Proposition 1. At last, in Section D, we present evaluation results on SVHN and CINIC10. We also conduct a study on the transferability of the coreset selected by CCS.

## B    DETAILED EXPERIMENTAL SETTING

**CIFAR10 and CIFAR100** (Krizhevsky et al., 2009). We use ResNet18 (He et al., 2016) as the network architecture for CIFAR10/CIFAR100. For all coresets with different pruning rates, we train models for $40,000$ iterations with a $256$ batch size (about $200$ epochs over the entire dataset). We use the SGD optimizer ($0.9$ momentum and $0.0002$ weight decay) with a $0.1$ initial learning rate. The learning rate scheduler is the cosine annealing learning rate scheduler (Loshchilov & Hutter, 2017) with a $0.0001$ minimum learning rate. We use a 4-pixel padding crop and a randomly horizontal flip as data augmentation.

**SVHN** (Netzer et al., 2011). We use ResNet18 (He et al., 2016) as the network architecture. For all coresets with different pruning rates, we train models for $30,000$ iterations with a $256$ batch size (about $100$ epochs over the entire dataset). We use the SGD optimizer ($0.9$ momentum and $0.0002$ weight decay) with a $0.1$ initial learning rate. The learning rate scheduler is the cosine annealing learning rate scheduler (Loshchilov & Hutter, 2017) with a $0.0001$ minimum learning rate.

**CINIC10** (Darlow et al., 2018). The CINIC10 dataset contains $270,000$ images in total and is evenly split to three subsets: training, valid, and test. Guided by (Darlow et al., 2018), we combine the training and valid set to form a large training dataset containing $180,000$ images and measure the test accuracy with the test set (containing $90,000$ examples). We use ResNet18 (He et al., 2016) as the network architecture. For all coresets with different pruning rates, we train models for $70,000$ iterations with a $256$ batch size (about $100$ epochs over the entire dataset). We use the SGD optimizer ($0.9$ momentum and $0.0002$ weight decay) with a $0.1$ initial learning rate. The learning rate scheduler is the cosine annealing learning rate scheduler (Loshchilov & Hutter, 2017) with a $0.0001$ minimum learning rate. We use a 4-pixel padding crop and a randomly horizontal flip as data augmentation.

**ImageNet** (Deng et al., 2009). We use ResNet34 (He et al., 2016) as the network architecture. For all coresets with different pruning rates, we train models for $300,000$ iterations with a $256$ batch size (about $60$ epochs over the entire dataset). We use the SGD optimizer ($0.9$ momentum and $0.0001$ weight decay) with a $0.1$ initial learning rate. The learning rate scheduler is the cosine annealing learning rate scheduler (Loshchilov & Hutter, 2017).

**Difficulty score calculation**. For CIFAR10 and CIFAR100, we train a model with the entire dataset for $200$ epochs to calculate the forgetting score and AUM and use the last checkpoint to calculate entropy for each training example. We use the first $10$ epoch's training information to calculate the EL2N score. For SVHN and CINIC10, we train a model with the entire dataset for $100$ epochs to calculate the forgetting score and AUM and use the last checkpoint to calculate entropy for each training example. For ImageNet, we train the entire dataset for $90$ epochs to calculate the forgetting and AUM score. We use the first $10$ epoch's training information to calculate the EL2N score.

**Coverage-centric methods setting**. We set the number of strata $k = 50$ for all datasets and pruning rates. We use the grid search with $0.1$ step size to find an optimal hard cutoff rate $\beta$ for different datasets and pruning rates. For each dataset, we list the optimal $\beta$ value for every $\alpha$ as follows in the format of tuple $(\alpha, \beta)$. For CIFAR10, the optimal setting is $(30\%, 0)$, $(50\%, 0)$, $(70\%, 10\%)$, $(80\%, 10\%)$, $(90\%, 30\%)$. For CIFAR100, the optimal setting is $(30\%, 10\%)$, $(50\%, 20\%)$, $(70\%, 20\%)$, $(80\%, 40\%)$, $(90\%, 50\%)$. For SVHN, the optimal setting is $(30\%, 0)$, $(50\%, 0)$, $(70\%, 10\%)$, $(90\%, 10\%)$, $(95\%, 20\%)$. For CINIC10, the optimal setting is $(30\%, 0)$, $(50\%, 0)$, $(70\%, 10\%)$, $(80\%, 10\%)$, $(90\%, 20\%)$. For ImageNet, the optimal setting is $(30\%, 0)$, $(50\%, 10\%)$, $(70\%, 20\%)$, $(80\%, 20\%)$, $(90\%, 30\%)$.

Each model is trained on a NVIDIA 2080TI GPU. We also include the implementation in the supplementary material.

## C  THEORETICAL ANALYSIS

We include full proofs of Theorem 1 and Proposition 1 in this section.

### C.1  PROOF OF THEOREM 1

Our proof flow is similar to the proof of Theorem in Sener & Savarese (2017).

**Lemma 1.** *Berlind & Urner (2015) Fix some $p, p' \in [0, 1]$ and $y' \in \{0, 1\}$. Then,*

$$p_{y \sim p}(y = y') \leq p_{y \sim p'}(y = y') + |p - p'| \tag{3}$$

**Theorem 1.** *Given n i.i.d. samples drawn from $P_\mu$ as $\mathcal{S} = \{x_i, y_i\}_{i \in [n]}$ where $y_i \in [C]$ is the class label for example $x_i$, a coreset $\mathcal{S}'$ which is a p-partial r-cover for $P_\mu$ on the input space $X$, and an $\epsilon > 1 - p$, if the loss function $l(\cdot, y, w)$ is $\lambda_l$-Lipschitz continuous for all $y$, $w$ and bounded by L, the class-specific regression function $\eta_c(x) = p(y = c|x)$ is $\lambda_\eta$-Lipschitz for all c, and $l(x, y; h_{\mathcal{S}'}) = 0$, $\forall (x, y) \in \mathcal{S}'$, then with probability at least $1 - \epsilon$:*

$$\left| \frac{1}{n} \sum_{x,y \in \mathcal{S}} l(x, y; h_{\mathcal{S}'}) \right| \leq r(\lambda_l + \lambda_\eta LC) + L\sqrt{\frac{\log \frac{p}{p+\epsilon-1}}{2n}}$$

*Proof.* We first bound $\mathbb{E}_{y_i \sim \eta(\mathbf{x}_i)}[l(\mathbf{x}_i, y_i; h_{\mathcal{S}'})]$, where $\eta(x)$ is the output of the model, and we represent $\{y_i = k\} \sim \eta_k(\mathbf{x}_i)$ with $y_i \sim \eta_k(\mathbf{x}_i)$. We have a condition which states that there exists an $\mathbf{x}_j$ in $\delta$ ball around $\mathbf{x}_i$ such that $\mathbf{x}_j$ has 0 loss. Since $\mathcal{S}'$ is a p-partial r-cover for the distribution, for $x_i \sim \mu$, with the probability p, there is $x_j \in \mathcal{S}'$ such that $|x_i - x_j| \leq r$ and $x_j$ has 0 loss.

The following inequality holds with the probability $p$:

$$E_{y_i \sim \eta(\mathbf{x}_i)}[l(\mathbf{x}_i, y_i; h_{\mathcal{S}'})]$$
$$= \sum_{k \in [C]} p_{y_i \sim \eta_k(\mathbf{x}_i)}(y_i = k) l(\mathbf{x}_i, k; h_{\mathcal{S}'})$$

By applying Lemma 1 and 0 training loss condition on $x_j$, we have:

$$\leq \sum_{k \in [C]} p_{y_i \sim \eta_k(\mathbf{x}_j)}(y_i = k) l(\mathbf{x}_i, k; h_{\mathcal{S}'}) + \sum_{k \in [C]} |\eta_k(\mathbf{x}_i) - \eta_k(\mathbf{x}_j)| l(\mathbf{x}_i, k; h_{\mathcal{S}'})$$
$$= \sum_{k \in [C]} p_{y_i \sim \eta_k(\mathbf{x}_j)}(y_i = k)[l(\mathbf{x}_i, k; h_{\mathcal{S}'}) - l(\mathbf{x}_j, k; h_{\mathcal{S}'})] + \sum_{k \in [C]} |\eta_k(\mathbf{x}_i) - \eta_k(\mathbf{x}_j)| l(\mathbf{x}_i, k; h_{\mathcal{S}'}).$$

By applying the Lipschitz property of regression function and loss function, we have:

$$\leq \sum_{k \in [C]} p_{y_i \sim \eta_k(\mathbf{x}_j)}(y_i = k) \lambda_l |x_i - x_j| + \sum_{k \in [C]} l(\mathbf{x}_i, k; h_{\mathcal{S}'}) \lambda_\eta |x_i - x_j|$$
$$\leq \lambda_l |x_i - x_j| + \sum_{k \in [C]} l(\mathbf{x}_i, k; h_{\mathcal{S}'}) \lambda_\eta |x_i - x_j|$$

By applying the loss bound and the p-partial r-cover constraint, we have:

$$\leq \lambda_l |x_i - x_j| + LC\lambda_\eta |x_i - x_j| = (\lambda_l + \lambda_\eta LC)|x_i - x_j|$$
$$\leq r(\lambda_l + \lambda_\eta LC).$$

At last, we have the following inequality with probability $p$:

$$E_{y_i \sim \eta(\mathbf{x}_i)}[l(\mathbf{x}_i, y_i; h_{\mathcal{S}'})] \leq r(\lambda_l + \lambda_\eta LC). \tag{4}$$

Also, by the Hoeffding's Bound (Hoeffding, 1994), with probability at least $1 - \gamma$, we have:

$$\left| \frac{1}{n} \sum_{x,y \in \mathcal{S}} l(x, y; h_{\mathcal{S}'}) - E_{y_i \sim \eta(\mathbf{x}_i)}[l(\mathbf{x}_i, y_i; h_{\mathcal{S}'})] \right| \leq L\sqrt{\frac{\log(1/\gamma)}{2n}}. \tag{5}$$

By combining Equation 4 and Equation 5, we have that with probability at least $1 - \epsilon$:

$$\left| \frac{1}{n} \sum_{\boldsymbol{x}, y \in \mathcal{S}} l(\boldsymbol{x}, y; h_{\mathcal{S}'}) \right| \leq r(\lambda_l + \lambda_\eta LC) + L \sqrt{\frac{\log \frac{p}{p+\epsilon-1}}{2n}}$$

with $p > 1 - \epsilon$. $\square$

## C.2 Proof of Proposition 1

**Proposition 1.** *Given a metric space $(X, d)$, a distribution $P_\mu$, cover percentage $p$, a set $\mathcal{S}$, and $f_r(p) : [0, 1] \rightarrow [0, +\infty]$ representing the mapping between $p$ and $r$, if $f_r$ is Riemann-integrable, the AUC of the p-r curve $AUC_{pr}(\mathcal{S}) = \int_0^1 f_r(p)dp = \mathbb{E}_{x \sim P_\mu}[d(S, \boldsymbol{x})]$, where $d(S, \boldsymbol{x}) = \min_{\boldsymbol{x}' \in S} d(\boldsymbol{x}', \boldsymbol{x})$.*

*Proof.* Since $f_r$ is Riemann-integrable, we can calculate $\int_0^1 f_r(p)dp$, which is $AUC_{pr}$, with Riemann sum. Specifically, by evenly dividing $[0, 1]$ to $0 = p_0 < p_1 < p_2 < ... < p_n = 1$, we have:

$$\int_0^1 f_r(p)dp = \lim_{n \to \infty} \sum_{i=1}^n f_r(p_i)(p_i - p_{i-1}).$$

Since $p_i = P_\mu(d(\mathcal{S}, x) \leq f_r(p_i))$, we have

$$\int_0^1 f_r(p)dp = \lim_{n \to \infty} \sum_{i=1}^n f_r(p_i) P_\mu(f_r(p_{i-1}) < d(\mathcal{S}, x) \leq f_r(p_i))$$

We know $f_r$ is an increasing function and $f_r(0) = 0$. Let $\Delta_i = f_r(p_i) - f_r(p_{i-1})$, we have:

$$= \lim_{n \to \infty} \sum_{i=1}^n (\sum_{j=1}^i \Delta_j) P_\mu(f_r(p_{i-1}) < d(\mathcal{S}, x) \leq f_r(p_{i-1}) + \Delta_i)$$

As $n \to \infty$, we have $\Delta_i \to 0$, so we have:

$$= \lim_{\Delta \to 0} \int_0^\infty r P_\mu(r \leq d(\mathcal{S}, \boldsymbol{x}) < r + \Delta) dr,$$

which is the expectation of the minimum distance to data in $\mathcal{S}$.

So we conclude that $AUC_{pr}(\mathcal{S})$ is the expectation of the minimum distance to data in $\mathcal{S}$.

$\square$

# D Additional evaluation results

## D.1 Evaluation results on SVHN and CINIC10

In this section, we present our comparison experiment on SVHN and CINIC10 in Table 5 and Table 6, respectively. Since SVHN classification is simpler than other classification tasks, we also study the performance of coreset with a $95\%$ pruning rate. We find similar comparison results that we discussed in Section 4.1. For both datasets, we observe that CCS achieves comparable accuracy with low pruning rates, but higher accuracy than random and other baseline coreset selection methods.

## D.2 Coreset coverage analysis

Figure 7 and Table 7 compare $AUC_{pr}$ between CCS with forgetting score and the vanilla forgetting methods (pruning easy first) on the CIFAR10 dataset. We observe that CCS has a lower $AUC_{pr}$ and better accuracy with the same pruning rate, which suggests that the proposed coverage-centric method does improve the data coverage of the coresets.

Table 3: We compare the testing accuracy of CCS against baseline methods and Random on CI-FAR10 dataset at different pruning rates. When we prune out 30% or 50% fraction of data, CCS achieves comparable accuracy. At a 70%, 80%, or 90% pruning rate, CCS outperforms baselines by a large margin. The model trained with the entire dataset has 95.23% accuracy.

| Pruning Rate | 30% | 50% | 70% | 80% | 90% |
|---|---|---|---|---|---|
| Random | $94.33_{\pm 0.17}$ | $93.4_{\pm 0.17}$ | $90.94_{\pm 0.38}$ | $87.98_{\pm 0.39}$ | $79.04_{\pm 1.53}$ |
| Entropy | $94.44_{\pm 0.2}$ | $92.11_{\pm 0.47}$ | $85.67_{\pm 0.71}$ | $79.08_{\pm 0.36}$ | $66.52_{\pm 1.08}$ |
| Forgetting | $95.36_{\pm 0.13}$ | $\mathbf{95.29_{\pm 0.18}}$ | $90.56_{\pm 1.8}$ | $62.74_{\pm 2.42}$ | $34.03_{\pm 1.05}$ |
| EL2N | $\mathbf{95.44_{\pm 0.15}}$ | $94.61_{\pm 0.20}$ | $87.48_{\pm 1.33}$ | $70.32_{\pm 2.11}$ | $22.33_{\pm 0.54}$ |
| AUM | $95.07_{\pm 0.24}$ | $95.26_{\pm 0.15}$ | $91.36_{\pm 1.4}$ | $57.84_{\pm 4.1}$ | $28.06_{\pm 1.09}$ |
| Moderate | $93.86_{\pm 0.11}$ | $92.58_{\pm 0.30}$ | $90.56_{\pm 0.27}$ | $87.32_{\pm 0.38}$ | $81.04_{\pm 1.63}$ |
| Forgetting (CCS) | $95.40_{\pm 0.12}$ | $95.04_{\pm 0.37}$ | $92.97_{\pm 0.25}$ | $90.93_{\pm 0.22}$ | $85.70_{\pm 0.36}$ |
| EL2N (CCS) | $94.84_{\pm 0.15}$ | $94.03_{\pm 0.24}$ | $92.25_{\pm 0.24}$ | $89.81_{\pm 0.30}$ | $\mathbf{86.68_{\pm 1.25}}$ |
| AUM (CCS) | $95.27_{\pm 0.06}$ | $94.93_{\pm 0.18}$ | $\mathbf{93.00_{\pm 0.16}}$ | $\mathbf{90.91_{\pm 0.27}}$ | $86.08_{\pm 0.61}$ |

Table 4: Accuracy results on CIFAR100 comparing CCS against other baselines. At low pruning rate, CCS achieves comparable accuracy to baselines. At higher pruning rates, CCS outperforms baselines and random. The model trained with the entire dataset has 78.21% accuracy.

| Pruning Rate | 30% | 50% | 70% | 80% | 90% |
|---|---|---|---|---|---|
| Random | $74.59_{\pm 0.27}$ | $71.07_{\pm 0.4}$ | $65.3_{\pm 0.21}$ | $57.36_{\pm 0.64}$ | $44.76_{\pm 1.58}$ |
| Entropy | $72.26_{\pm 0.08}$ | $63.26_{\pm 0.29}$ | $50.49_{\pm 0.88}$ | $41.83_{\pm 0.33}$ | $28.96_{\pm 0.78}$ |
| Forgetting | $76.91_{\pm 0.32}$ | $68.6_{\pm 1.02}$ | $38.06_{\pm 1.14}$ | $24.23_{\pm 0.59}$ | $15.93_{\pm 0.24}$ |
| EL2N | $76.25_{\pm 0.24}$ | $65.90_{\pm 1.06}$ | $34.42_{\pm 1.50}$ | $15.51_{\pm 1.20}$ | $8.36_{\pm 0.19}$ |
| AUM | $76.93_{\pm 0.32}$ | $67.42_{\pm 0.49}$ | $30.64_{\pm 0.58}$ | $16.38_{\pm 0.4}$ | $8.77_{\pm 0.35}$ |
| Moderate | $74.60_{\pm 0.41}$ | $71.10_{\pm 0.24}$ | $65.34_{\pm 0.41}$ | $58.51_{\pm 0.84}$ | $45.51_{\pm 1.26}$ |
| Forgetting (CCS) | $\mathbf{77.14_{\pm 0.31}}$ | $\mathbf{74.45_{\pm 0.16}}$ | $\mathbf{68.92_{\pm 0.12}}$ | $\mathbf{63.99_{\pm 0.37}}$ | $\mathbf{55.59_{\pm 0.7}}$ |
| EL2N (CCS) | $75.02_{\pm 0.12}$ | $72.09_{\pm 0.53}$ | $67.13_{\pm 0.22}$ | $61.83_{\pm 0.93}$ | $52.55_{\pm 1.63}$ |
| AUM (CCS) | $76.84_{\pm 0.25}$ | $73.77_{\pm 0.21}$ | $68.85_{\pm 0.08}$ | $63.2_{\pm 0.54}$ | $55.03_{\pm 0.72}$ |

## D.3 TRANSFERABILITY OF CORESETS

Similar to Coleman et al. (2019), we also conduct a coreset transferability study. We train a ResNet18 model with the entire CIFAR10 dataset and use the forgetting score to select coresets with different pruning rates. Then we train the ResNet50 model with these coresets. The evaluation results are shown in Table 8. We find that coresets show good transferability across these two architectures.

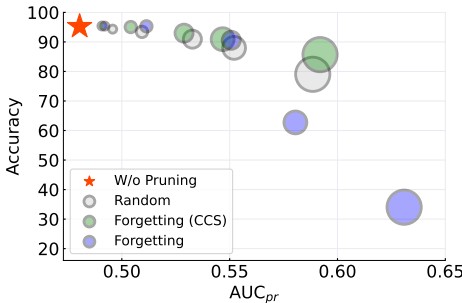

Figure 7: The relationship between $AUC_{pr}$ and accuracy. A larger circle size indicates a higher pruning ratio on the dataset. Compared to the vanilla forgetting method, CCS with the forgetting score achieves a lower $AUC_{pr}$ and better accuracy.

Table 5: Accuracy results on SVHN comparing CCS against other baseline methods. Pruning rate is the fraction of examples that are removed from the original dataset. We report the testing accuracy of the model trained on the pruned dataset. The model trained with the entire dataset has 96.12% accuracy. When we prune out 30% or 50% fraction of data, CCS achieves comparable accuracy. With a 70%, 90%, or 95% pruning rate, CCS outperforms baseline methods by a large margin.

| Pruning Rate | 30% | 50% | 70% | 90% | 95% |
|---|---|---|---|---|---|
| Random | $95.38_{\pm 0.4}$ | $94.81_{\pm 0.24}$ | $93.39_{\pm 0.27}$ | $89.58_{\pm 0.99}$ | $85.34_{\pm 1.01}$ |
| Entropy | $95.47_{\pm 0.45}$ | $94.95_{\pm 0.14}$ | $93.14_{\pm 0.55}$ | $81.6_{\pm 1.65}$ | $54.6_{\pm 3.88}$ |
| Forgetting | $95.57_{\pm 0.27}$ | $95.37_{\pm 0.29}$ | $94.76_{\pm 0.09}$ | $66.57_{\pm 2.83}$ | $27.95_{\pm 1.21}$ |
| EL2N | $\mathbf{96.05_{\pm 0.13}}$ | $95.61_{\pm 0.25}$ | $94.87_{\pm 0.28}$ | $56.53_{\pm 1.68}$ | $21.05_{\pm 1.85}$ |
| AUM | $95.91_{\pm 0.38}$ | $\mathbf{95.76_{\pm 0.10}}$ | $\mathbf{94.93_{\pm 0.17}}$ | $52.58_{\pm 5.98}$ | $21.61_{\pm 1.68}$ |
| Moderate | $93.91_{\pm 0.68}$ | $93.06_{\pm 0.30}$ | $92.74_{\pm 0.88}$ | $89.10_{\pm 0.28}$ | $85.66_{\pm 0.40}$ |
| Forgetting (CCS) | $95.96_{\pm 0.1}$ | $95.5_{\pm 0.15}$ | $94.56_{\pm 0.24}$ | $91.49_{\pm 0.89}$ | $87.78_{\pm 0.8}$ |
| EL2N (CCS) | $95.42_{\pm 0.16}$ | $95.59_{\pm 0.17}$ | $95.05_{\pm 0.14}$ | $92_{\pm 0.60}$ | $88.55_{\pm 0.78}$ |
| AUM (CCS) | $96.02_{\pm 0.19}$ | $95.69_{\pm 0.18}$ | $94.88_{\pm 0.41}$ | $\mathbf{92.27_{\pm 0.67}}$ | $\mathbf{88.86_{\pm 1.11}}$ |

Table 6: Accuracy results on CINIC10 comparing CCS against other baseline methods. Pruning rate is the fraction of examples that are removed from the original dataset. We report the testing accuracy of the model trained on the pruned dataset. The model trained with the entire dataset has 89.97% accuracy. When we prune out 30% or 50% fraction of data, CCS achieves comparable accuracy. With a 70%, 80%, or 90% pruning rate, CCS outperforms baseline methods by a large margin.

| Pruning Rate | 30% | 50% | 70% | 80% | 90% |
|---|---|---|---|---|---|
| Random | $88.88_{\pm 0.07}$ | $87.64_{\pm 0.11}$ | $85.13_{\pm 0.17}$ | $82.63_{\pm 0.14}$ | $77.31_{\pm 0.66}$ |
| Entropy | $89.08_{\pm 0.11}$ | $86.69_{\pm 0.21}$ | $82.1_{\pm 0.15}$ | $77.37_{\pm 0.51}$ | $67.21_{\pm 0.89}$ |
| Forgetting | $90.07_{\pm 0.11}$ | $\mathbf{89.68_{\pm 0.16}}$ | $81.96_{\pm 1.07}$ | $61.26_{\pm 1.85}$ | $45.06_{\pm 1.48}$ |
| EL2N | $89.93_{\pm 0.11}$ | $87.65_{\pm 0.25}$ | $65.10_{\pm 0.71}$ | $36.09_{\pm 1.37}$ | $18.50_{\pm 0.25}$ |
| AUM | $\mathbf{90.11_{\pm 0.04}}$ | $89.5_{\pm 0.16}$ | $66.23_{\pm 0.69}$ | $33.89_{\pm 0.7}$ | $18.61_{\pm 0.15}$ |
| Moderate | $87.25_{\pm 0.11}$ | $86.04_{\pm 0.08}$ | $84.15_{\pm 0.10}$ | $82.09_{\pm 0.2}$ | $78.10_{\pm 0.31}$ |
| Forgetting (CCS) | $90.03_{\pm 0.14}$ | $89.47_{\pm 0.08}$ | $\mathbf{88.04_{\pm 0.04}}$ | $85.88_{\pm 0.20}$ | $81.04_{\pm 0.94}$ |
| EL2N (CCS) | $89.52_{\pm 0.11}$ | $88.41_{\pm 0.17}$ | $87.76_{\pm 0.12}$ | $\mathbf{86.20_{\pm 0.17}}$ | $\mathbf{83.18_{\pm 0.56}}$ |
| AUM (CCS) | $90.04_{\pm 0.09}$ | $89.24_{\pm 0.11}$ | $87.99_{\pm 0.04}$ | $85.82_{\pm 0.27}$ | $81.92_{\pm 0.44}$ |

Table 7: $\text{AUC}_{pr}$ comparison between the vanilla forgetting method and CCS. Coresets obtained by CCS show a lower $\text{AUC}_{pr}$, which suggests better data coverage.

| Pruning Rate | 30% | 50% | 70% | 80% | 90% |
|---|---|---|---|---|---|
| Forgetting | 0.492 | 0.511 | 0.551 | 0.580 | 0.631 |
| Forgetting (CCS) | 0.491 | 0.504 | 0.529 | 0.547 | 0.592 |

Table 8: We train ResNet50 with coresets of CIFAR10 selected by the forgetting score calculated on training information with ResNet18. The evaluation results shows that the coresets selected by CCS have good transferability across models.

| Pruning Rate | 30% | 50% | 70% | 80% | 90% |
|---|---|---|---|---|---|
| Random | 94.53 | 92.87 | 89.03 | 88.48 | 78.94 |
| Forgetting | 95.22 | 94.67 | 91.95 | 71.10 | 35.21 |
| Forgetting (CCS) | 95.08 | 94.99 | 92.84 | 89.59 | 85.65 |

## D.4 ADDITIONAL EVALUATION ON IMAGENET

In this part, we present our comparison experiment on ImageNet, a larger scale dataset compared to the other four datasets evaluated, in Table 9. Due to the large computational cost for the ImageNet training, we only apply CCS on the AUM importance score, and similar to the prior work Sorscher et al. (2022), we train each model for one time.

From Table 9, we notice that CCS outperforms all baseline methods at high pruning rates and achieves comparable performance at low pruning rates. We find that EL2N and AUM still experience the catastrophic accuracy drop at high pruning rates. Specifically, compared to random sampling, CCS achieves a $5.02\%$ improvements at a $90\%$ pruning rate. The evaluation results on ImageNet indicate that CCS is also scaled to the large dataset.

Forgetting performs very similarly to random at high pruning rates. We think the reason behind this is that the training dynamics of ImageNet are only collected for 90 epochs, which causes an example to be forgotten at most 45 times. Considering the huge number of the ImageNet dataset, many example can share the same forgetting number. Since coreset selection will randomly sample examples when the scores of these examples are the same, forgetting method for ImageNet adds more randomness to the coreset selection.

Table 9: Accuracy results on ImageNet comparing CCS against other baseline methods. Pruning rate is the fraction of examples that are removed from the original dataset. We report the testing accuracy of the model trained on the pruned dataset. The model trained with the entire dataset has $73.54\%$ Top-1 accuracy. When we prune out $30\%$ or $50\%$ fraction of data, CCS achieves comparable accuracy. With a $70\%$, $80\%$, or $90\%$ pruning rate, CCS outperforms baseline methods by a large margin.

| Pruning Rate | 30% | 50% | 70% | 80% | 90% |
|---|---|---|---|---|---|
| Random | 72.18 | 70.34 | 66.67 | 62.54 | 52.34 |
| Entropy | 72.34 | 70.76 | 64.04 | 55.8 | 39.04 |
| Forgetting | **72.6** | **70.89** | 66.51 | 62.92 | 52.28 |
| EL2N | 72.2 | 67.17 | 48.79 | 31.22 | 12.99 |
| AUM | 72.53 | 66.57 | 40.42 | 21.12 | 9.93 |
| Moderate | 72.06 | 70.32 | 65.86 | 61.3 | 52.16 |
| AUM (CCS) | 72.29 | 70.52 | **67.78** | **64.47** | **57.36** |

## D.5 DISCUSSION ON IMPORTANCE SAMPLING BASED METHOD

Another potential way to resolve the coverage issue is the importance sampling based method: sample each example in the data set with a probability based to its importance score. Examples with less importance are assigned smaller sampling probability and examples with larger importance are assigned larger sampling probability. We present preliminary experiments on CIFAR10, the AUM as the underlying importance metric, and using a softmax function to map importance scores to probabilities.

Unfortunately, experimental results in Table 10 show that the above scheme failed to beat random at high pruning rates. It did beat the SOTA method at high pruning rates. Not beating random is a big deal, since random selection is the simplest possible strategy for constructing a coreset.

The above result also suggests that *it is not easy to design a coreset selection scheme that beats random selection at high pruning rates.* Our proposed scheme, CCS does that on five of the datasets that we evaluated it on, and it is the first coreset selection scheme to achieve that, as far as we are aware.

We note that the relationship between data importance score and good-property sampling probability is not obvious. Unlike many statistical machine learning methods like KNN or k-means, which usually have a good math property, data importance scores in deep learning classification are usually heuristically calculated (as we discussed at the beginning of Section 4), which makes it non-trivial to figure out an effective way to calculate probabilities based on heuristic importance scores.

Table 10: Accuracy results on CIFAR10 comparing importance sampling against other baseline methods. Pruning rate is the fraction of examples that are removed from the original dataset. We report the testing accuracy of the model trained on the pruned dataset. Importance sampling shows a better performance than AUM at high pruning rates, but it has a worse performance than CCS and random sampling. We highlight the highest accuracy for each pruning rate in bold.

| Pruning Rate | 30% | 50% | 70% | 80% | 90% |
|---|---|---|---|---|---|
| Random | 94.33 | 93.40 | 90.94 | 87.98 | 79.04 |
| AUM | 95.07 | **95.26** | 91.36 | 57.84 | 28.06 |
| AUM (CCS) | **95.27** | 94.93 | **93.00** | **90.91** | **86.08** |
| AUM (Importance sampling) | 93.12 | 92.61 | 90.77 | 86.27 | 77.66 |

For the results in Table 10, we used the popular softmax function. More specifically, since a smaller AUM score indicates a larger importance, we calculated the sampling probability $p_i$ for the $i$th example as follows:

$$p_i = \frac{e^{\frac{s_{max} - s_i}{s_{max}}}}{\sum_{n=1}^{N} e^{\frac{s_{max} - s_n}{s_{max}}}},$$

where $s_i$ is the AUM score for each example, $s_{max}$ is the maximum possible AUM score, and $N$ is the number of examples.

We do not rule out that a better-designed importance sampling technique exists, say, by rethinking the relationship between importance score and probability of selection. CCS uses stratified sampling, which we found to work really well. CCS thus provides a good baseline for any further work in this area.

