# OpenReview forum: "Coverage-centric Coreset Selection for High Pruning Rates"
_ICLR.cc/2023/Conference — ICLR 2023 poster_

### Official Review · Reviewer_6QHN · 2022-10-22

**Confidence:** 3
**Correctness:** 3
**Technical Novelty And Significance:** 2
**Empirical Novelty And Significance:** 3
**Recommendation:** 6

**Clarity, Quality, Novelty And Reproducibility:**

# Clarity

The paper did a comprehensive comparison to directly related works. Key assumptions and choice of parameters are discussed, and sufficient technical details have been provided. However, as mentioned in "Weakness", the comparison to the study of coreset and importance sampling techniques in general is poorly covered.

Minor comments:

1. Section 4, paragraph "Baselines". If I understand correctly, many of the listed algorithms are not really "baselines", since they define how the importance score is picked which is your input, instead of how the coreset is chosen which is your algorithm. Please clarify.

2. In the description of Algorithm 1, it's better to state that \beta \leq 1 - \alpha (if I get it correctly).

3. In Section 2.1, you used "b" to denote the size of the coreset. However, in other places we are mostly talking about "pruning rate". What about consistently mentioning one of these, throughout the paper?

# Quality

The empirical improvement over baselines seem to be significant. The writing quality is excellent. The claims are grounded by either theoretical or empirical analysis. However, the theoretical analysis, Theorem 1, seems to be weak, particularly that it is only about the "ideal" case which the algorithm may not achieve. The (worst-case) guarantee of the algorithm is not analyzed at all.

# Originality

The main novelty is a new sufficient condition for a good coreset (which can also be used to evaluate a coreset selection method empirically), plus a new sampling scheme that outperforms existing results in high pruning rate. However, both are mostly based on existing methods, and I don't find it requires significant modifications. But indeed, the exact combination of techniques may be nontrivial to discover, especially considering that it performs well in empirical evaluations.

**Strength And Weaknesses:**

# Strength:

The technical idea and the reasoning is well grounded by theoretical and empirical analysis, and the comparison with previous methods seem to be comprehensive, which is convincing. The new algorithm makes sense, and the empirical result shows a significant improvement over previous results.

# Weakness:

The reasoning generally makes sense, but I do find a few important aspects not discussed (I might be wrong since I mostly work on coresets for clustering instead of classification). In particular:

1. From the literature of coresets for clustering, once importance score is given, the next step is usually importance sampling: sample each point x in the data set with probability *proportional to its importance score*. Note that this distribution can sample both easiest and hardest points, and more importantly, if the easy points are a lot, each of them is sampled with small probability but the combined probability of sampling an easy point can be very significant. Hence, this seems to resolve the issue that you mention, where easy points tend to be very “dense”, and simply focusing on hard points may miss them. Unfortunately, I don't see this very natural approach discussed/reflected in the paper.

2. Actually, I don’t find any reference about coresets for clustering (and related problems), especially those using importance sampling. Consider to cite and discuss the typical ones, such as

* Turning Big Data Into Tiny Data: Constant-Size Coresets for k-Means, PCA, and Projective Clustering, Feldman et al, SICOMP 20.

* A unified framework for approximating and clustering data, Feldman and Langberg, STOC 11.

3. The entire paper seems to assume the importance score is given which can be arbitrary. However, the quality of the importance score can also greatly affect the performance, but this does not seem to be modeled/considered in the algorithm? Moreover, I find it unclear whether improving the sampling algorithm or improving the importance score is the most significant issue of the existing algorithms. For instance, it is a fact that the previous works perform worse than random sampling at 90% pruning rate, but why this has to be caused by a bad sampling algorithm? What about using an improved importance score?

**Summary Of The Paper:**

This paper gives an improved one-shot coreset selection method for classification problems. A one-shot coreset is a subset of the training set, and the goal is to find a coreset/subset of a given size such that the training error on the coreset is minimized.

The paper first studies the ability of the coreset to “cover” the data distribution, and justifies why this is important. To this end, the notion of partial cover was proposed. Roughly, a subset S is a p-partial r-cover for some distribution, if the total probability measure of the union of all radius-r neighborhoods around each point in S is p. This generalizes a previous notion which is essentially p = 1. In Theorem 1, this paper proves that if a coreset is a p-partial r-cover, then it has a bounded error w.h.p. on the training set (where the randomness comes from the training set). Hence, this is a valid error measure. To further stabilize the measure, the AUC (area under curve) measure is considered, which may be viewed as an average/overall measure of the radius r, over all cover percentage.

The paper observes that the AUC of several existing coreset algorithms is worse than a naive uniform sampling, when the pruning rate is high (i.e., 90%). The focus is thus to propose a new algorithm that improves the performance when the pruning rate is high. The key observation is that the previous results drop points too aggressively under high pruning rate. Hence, given a set of importance score of data points, the new algorithm, called CCS, first excludes a \beta fraction (which is smaller than the total pruning rate) of the hardest examples (according to the score), and then picks the data points to be kept in a somewhat uniform way.

Experiment results show that the proposed method has a significant improvement over previous coreset methods when the pruning rate is high, regardless how the importance score is defined (tested on at least 4 different scoring schemes), while still matches, or slightly outperforms, the existing methods when the pruning rate is low.

**Summary Of The Review:**

This is a nice paper. However, I would suggest a weak reject for now, due to the concerns that I raised in the "Weakness" part. They can significantly affect the justification of novelty of the paper.

---

> ### Author Response · Authors · 2022-11-13
> **Response for Reviewer 6QHN (Part 1)**
>
> Thank you for your constructive review! We address your questions and concerns below:
>
> >1. From the literature of coresets for clustering, once importance score is given, the next step is usually importance sampling: sample each point x in the data set with probability proportional to its importance score. Note that this distribution can sample both easiest and hardest points, and more importantly, if the easy points are a lot, each of them is sampled with small probability but the combined probability of sampling an easy point can be very significant. Hence, this seems to resolve the issue that you mention, where easy points tend to be very “dense”, and simply focusing on hard points may miss them. Unfortunately, I don't see this very natural approach discussed/reflected in the paper.
>
> Thank you for the suggestion. We agree with the reviewer that probability-based importance sampling which is commonly used in coreset for clustering is a potential alternative to address the coverage issue in coreset selection for deep learning. We decided to evaluate proportional importance sampling on CIFAR10. We mapped importance scores to probabilities using a softmax function and then picked examples based on their probabilities to form the coreset.
>
> Unfortunately, the proportional importance sampling method underperforms random sampling at high pruning rates; see the AUM (Importance Sampling) row in the table below.  It performs better than a SOTA method (AUM) at high pruning rates,  but underperforms random sampling. Beating random selection is crucial for any coreset selection algorithm since random selection is the simplest selection strategy. We add these evaluation details on the proportional importance sampling in Appendix D.5 and mention it in the Evaluation section.
>
>
> | Pruning rate              | 30%   | 50%   | 70%   | 80%   | 90%   |
> |---------------------------|-------|-------|-------|-------|-------|
> | Random                    | 94.33 | 93.40 | 90.94 | 87.98 | 79.04 |
> | AUM                       | 95.07 | 95.26 | 91.36 | 57.84 | 28.06 |
> | AUM (CCS)         | 95.27 | 94.93 | 93.00 | 90.91 | 86.08 |
> | AUM (Importance Sampling) | 93.12 | 92.61 | 90.77 | 86.27 | 77.60  |
>
> *The above experiment results suggest that it is non-trivial to beat random sampling at high pruning rates, even with a scheme that improves data coverage.* Earlier in our work, we ourselves wondered whether any scheme could beat random at high pruning rates since all existing SOTA methods were underperforming random selection at high pruning rates.
>
> CCS achieves that remarkable result  -- it beats random consistently for five datasets at high pruning rates and also beats SOTA methods by an even larger margin. We thus believe that CCS would be a valuable contribution to the research community.
>
> We do not rule out that there could be other ways to map importance scores to probabilities that cause  proportional importance sampling to perform better.  But,  CCS based on stratified sampling performs really well. CCS thus provides a good baseline for future work in this direction.
>
> >2. Actually, I don’t find anyreference about coresets for clustering (and related problems), especially those using importance sampling. Consider to cite and discuss the typical ones, such as: 1) Turning Big Data Into Tiny Data: Constant-Size Coresets for k-Means, PCA, and Projective Clustering, Feldman et al, SICOMP 20. 2) A unified framework for approximating and clustering data, Feldman and Langberg, STOC 11.
>
> Thank you for suggesting missing related work in the clustering field. We have updated the Related Work discussion accordingly in Section 5 to include and discuss these references.

---

> > ### Author Response · Authors · 2022-11-13
> > **Response for Reviewer 6QHN (Part 2)**
> >
> > >3. The entire paper seems to assume the importance score is given which can be arbitrary. However, the quality of the importance score can also greatly affect the performance, but this does not seem to be modeled/considered in the algorithm?
> >
> > We agree with the reviewer that the importance score influences the performance of CCS.  For instance, if the importance score were to be completely random, then CCS would be like random sampling.
> >
> > The design of a good importance score metric has been a subject of exploration in prior work on coreset selection. For example, Forgetting score, EL2N, and AUM are popular importance scores in this area and they all aim to find examples that are difficult to learn and thus provide more information. CCS is designed to be **complementary** to various importance scores and can be combined with them. Our evaluation results show that CCS achieves good performance when the importance scores effectively quantify data difficulty.
> >
> > > Moreover, I find it unclear whether improving the sampling algorithm or improving the importance score is the most significant issue of the existing algorithms. For instance, it is a fact that the previous works perform worse than random sampling at 90% pruning rate, but why this has to be caused by a bad sampling algorithm? What about using an improved importance score?
> >
> > The results in our paper suggest the former, i.e., improving the sampling algorithm appears more important than improving the importance score.  We evaluated three variants of CCS by combining it with three different importance metrics: AUM, Forgetting score, and EL2N. We found that all three CCS variants perform similarly, suggesting that CCS's performance is not very sensitive to the choice of importance score metric, as long as a reasonable metric is chosen that characterizes examples by difficulty.
> >
> > Conversely, CCS significantly outperforms SOTA sampling methods that were based on the same importance metric. That strongly suggests that the choice of sampling algorithm has a significant impact on the quality of the coreset for a given importance metric. Appendix D.5 provides further evidence that the sampling algorithm is really important.  Proportional importance sampling did not perform as well as CCS for the same importance metric.
> >
> > >1. Section 4, paragraph "Baselines". If I understand correctly, many of the listed algorithms are not really "baselines", since they define how the importance score is picked which is your input, instead of how the coreset is chosen which is your algorithm. Please clarify.
> >
> > We thank the reviewer for pointing this out. We agree that papers on which baselines are based define different importance metrics, but all these papers assume that the most important examples are to be selected first when designing a coreset, and the experimental results they present assume that as well. Thus, our baseline methods follow the underlying assumption in these papers to “prune easy examples first”, with the methods differing only on how they measure importance. We have revised Section 4 to clarify this.
> >
> > >2. In the description of Algorithm 1, it's better to state that \beta \leq 1 - \alpha (if I get it correctly).
> >
> > Thank you for pointing out this condition of Algorithm 1 to us. We have now added the condition $\beta \leq 1 - \alpha$ to Algorithm 1 to make the description more accurate.
> >
> > >3. In Section 2.1, you used "b" to denote the size of the coreset. However, in other places we are mostly talking about "pruning rate". What about consistently mentioning one of these, throughout the paper?
> >
> > Thanks for the suggestion. We made the change to the optimization problem in Section 2.1 to make it consistent throughout the paper.
> >
> > ---
> >
> > We thank the reviewer again for the constructive review and valuable comments.  Please let us know if we can provide more information to better address your concerns. We hope you can kindly consider updating the rating positively once you feel satisfied with the changes.

---

> ### Author Response · Authors · 2022-12-01
> **Thanks to Reviewer 6QHN and Response Summary**
>
> Dear Reviewer 6QHN,
>
> We greatly appreciate your time in reviewing our paper again. We would like to kindly remind you that the response and the revised paper are ready.
>
> *One of the reviewers replied that our responses successfully addressed the concerns and confusion and believes the paper to be above the threshold for acceptance.* We hope our responses can also address your concerns. We are looking forward to your feedback!
>
> **We would like to briefly summarize the response to your major concerns for your convenience.**
>
> > Importance sampling can be a more direct solution to handle the accuracy drop issue.
>
> We added an additional evaluation that finds that a potential implementation of importance sampling is not effective for addressing the accuracy drop issue. We agree with the reviewer that importance sampling is a promising mitigation strategy, but it is no-trivial to design.
>
> We also would like to note that, besides the baselines discussed in our paper, *our method also outperforms another submission to ICLR2023*, which receives quite positive ratings (https://openreview.net/forum?id=7D5EECbOaf9), which implies that our method design is not trivial or obvious.
>
> > Missing reference on the coreset for clustering.
>
> We thank the reviewer for pointing out missing related work; we already included and discussed those related work in the revision.
>
> ---
>
> We have already revised the paper to include these points in the paper. We also kindly refer you to our previous responses (both separate and general responses) for more detailed discussions and responses to your other concerns. Please let us know if we can provide more information to better address your concerns and clarify the claim in the paper!

---

### Official Review · Reviewer_17rD · 2022-10-24

**Confidence:** 4
**Correctness:** 4
**Technical Novelty And Significance:** 3
**Empirical Novelty And Significance:** 4
**Recommendation:** 6

**Clarity, Quality, Novelty And Reproducibility:**

Clarity: the hypothesis and the metrics are well-defined, but I find the CCS method needlessly complicated. Also my understanding of Algorithm 1 does not align with how it is portrayed in Figure 4.

Quality: the paper is of high quality, the reasoning and the metrics as well.

Novelty: the analysis and the fix are both novel, and the improvements are quite large.

Reproducibility: the code is provided, which is great (I did not test it for myself, though).

**Strength And Weaknesses:**

Strengths:
- The argument is well-grounded: it starts with an observation, then a hypothesis is presented, a metric is proposed to validate the hypothesis, and finally a method is proposed that empirically fixes the issue
- The improvements are quite large: the proposed method not only improves the SoTA at high-pruning rates, it beats the random baseline, which has been very strong at high-pruning rates until now, by a large margin

Weaknesses:
- The final method seems hacky and looking more closely, I'm not convinced that it does address the "coverage" issue directly. I can see how this method is way better at getting a distribution with higher coverage than the existing "hard cut-off" method. Anyway, maybe I'm expecting too much from the paper
- Relatedly, the method requires some hyper-parameter tuning, but the authors show that it's mostly invariant to one parameter and provide some intuition as to how to tune the other
- The method is only tested on datasets that are both small and fairly similar. Having results on ImageNet would make the results way more convincing.

Questions:
- The way I see it and by looking at Figure 2, AUC_pr metric might be dominated by the "r" values of the high coverage (high "p") regions. Do you think some normalization could be applied here to remove this bias or is that not necessary?
- I'm totally confused about the way CCS is represented in Figure 2. Looking at Algorithm 1, I'd expect CCS to have some a peak on the right side (i.e. hard cases that are removed on line 2 of Alg. 1) and then I'd expect CCS to have a more-or-less uniform distribution for the rest of the data (i.e. to be a scaled version of "all data") as you claim to take samples from all strata, not just a few. I'm not sure what I'm missing here.
- You show that the value of k is not important, but only show it for k>25. If you set k=1 and beta=0, do you recover random sampling? Because if you claim that k is not important, the only difference with random sampling would be just that you prune the hard samples. Is that correct?

**Summary Of The Paper:**

The authors study a known shortcoming of the SoTA coreset selection methods, which they call "catastrophic accuracy drop". The SoTA methods usually perform well under low-pruning regimes, but at high pruning rates they work even worse than random sampling.

The authors hypothesize that this is due to reduction in "data coverage" at high pruning rates. As such, they define "p-partial r-cover" for a selected subset and based on it propose the AUC_pr metric to quantify coverage. Then using this new metric they show that the SoTA methods indeed suffer from low data coverage at high pruning rates and AUC_pr of a selected subset correlates well with the performance of the models trained on it.

Finally, they propose a method, called Coverage-centric Coreset Selection (CCS), which uses stratified sampling over a given scoring mechanism. This method can complement the existing score-based pruning methods. They then experimentally show that their method can highly improve the existing SoTA methods at high pruning rates, both based on their proposed AUC_pr metric and the actual final model performance.

**Summary Of The Review:**

The authors' argument and observation seems quite valid and well-supported, but I'm confused by their method. I believe the AUC_pr metric could be quite valuable to the community, either in its current form, or an improved version of it. It will allow further research to take into account an important aspect that has not been easily addressable before.
As for the method, the improvements are quite large, but I have a hard time figuring out how the stratified sampling part is different from normal random sampling. I would've also like to see some evaluation of the method on a dataset such as ImageNet.

The main concerns that I have are that: 1) the proposed method seems a bit complicated/hacky and doesn't directly address the issues that are raised; and 2) I see some conflicts between Figure 4 and Alg 1.
3) I believe the metric might also need some adjusting in the future, but currently it should be adequate.

Whatever my reservations about the CCS method itself, the results are better than I expected.

---

> ### Author Response · Authors · 2022-11-13
> **Response for Reviewer 17rD (Part 1)**
>
> Thank you for your constructive review! We address your questions and concerns below:
>
> > The final method seems hacky and looking more closely, I'm not convinced that it does address the "coverage" issue directly. I can see how this method is way better at getting a distribution with higher coverage than the existing "hard cut-off" method. Anyway, maybe I'm expecting too much from the paper.
>
> We thank the reviewer for giving us an opportunity to address this concern.
>
> The insight for the method in part came from thinking about the problem from a theoretical perspective, which then helped us choose the $AUC{_pr}$ metric to characterize coverage. The final method evolved out of that when we realized that stratified sampling, with long roots in statistics, is a potential solution to improve data coverage and the $AUC{_pr}$ metric. We updated Section 3.3 to better describe the linkage between the metric and our sampling method.
>
> Our final method is really built on two simple insights: (1) really hard examples can detract from learning and one should consider pruning them; (2) stratified sampling is superior to picking just hard examples as in SOTA methods and merits evaluation.
>
> Both insights make sense when one thinks about the problem.  For the first insight, hard examples include mislabeled examples and they certainly hurt learning. Also, pruning hard examples allocates more budget to the high-density area to achieve better data coverage. Thus, considering pruning hard examples makes sense.
>
> The second insight says that learning only on hard examples does not necessarily imply that a model will learn to classify simpler examples well when decision boundaries are complex. For instance, catastrophic accuracy drop suggests that the model may be overfitting to hard examples at high pruning rates and failing to accurately classify simpler and more frequently-occurring examples. Stratified sampling addresses that.
>
> When the method is viewed from the above perspectives, we hope the reviewer finds the method less hacky and see how it addresses the coverage issue fairly directly, especially after we added an update to Section 3.3 to describe the intuition behind the proposed algorithm.
>
> >The method is only tested on datasets that are both small and fairly similar. Having results on ImageNet would make the results way more convincing.
>
> We agree with the reviewer that evaluation on ImageNet can better support our claim and make our argument more convincing. We now added the evaluation results on ImageNet. Here are the results for AUM metric on ImageNet:
>
> | Pruning rate | 30%   | 50%   | 70%   | 80%   | 90%   |
> |--------------|-------|-------|-------|-------|-------|
> | Random       | 72.18 | 70.34 | 66.67 | 62.54 | 52.34 |
> | AUM          | 72.53 | 66.57 | 40.42 | 21.12 | 9.93  |
> | AUM (CCS)    | 72.29 | 70.52 | 67.78 | 64.47 | 57.36 |
>
>
> The evaluation results show that our method, CCS, outperforms other methods at high pruning rates on the Imagenet dataset. The finding on ImageNet is consistent with findings on CIFAR10, CIFAR100, SVHN, and CINIC10.
>
> We include more ImageNet evaluation results in Appendix D.4.
>
>
> > The way I see it and by looking at Figure 2, AUC_pr metric might be dominated by the "r" values of the high coverage (high "p") regions. Do you think some normalization could be applied here to remove this bias or is that not necessary?
>
> This is a good observation. We also observed that high “r” examples may dominate the $AUC_{pr}$ metric. To eliminate the influence of these examples, when we calculate the $AUC_{pr}$ metric, we ignore the data misclassified by the model trained with the entire CIFAR10 dataset (which is mentioned in footnote 3), since these misclassified examples usually have a large $r$ value. That provides a form of mitigation to remove bias from high “r” examples. But, we agree with the reviewer that other forms of normalization may help improve the coverage measurement with the metric and are worthy of future research.
>
> > You show that the value of k is not important, but only show it for k>25. If you set k=1 and beta=0, do you recover random sampling? Because if you claim that k is not important, the only difference with random sampling would be just that you prune the hard samples. Is that correct?
>
> The reviewer’s observation is correct. Random sampling is a special case for our proposed method at k = 1 and beta = 0. (This also shows that with a proper choice of hyperparameters,  the performance of CCS can always be tuned to be no worse than random sampling.)   We have added this to the paper in Section 4.2.
>
> Regarding the sensitivity to $k$, we intended to say that when $k$ is reasonably large, the number of strata is not a sensitive hyperparameter. We have revised the text in Section 4.2 of the paper to better clarify this point.

---

> > ### Author Response · Authors · 2022-11-13
> > **Response for Reviewer 17rD (Part 2)**
> >
> > > I'm totally confused about the way CCS is represented in Figure 2. Looking at Algorithm 1, I'd expect CCS to have some a peak on the right side (i.e. hard cases that are removed on line 2 of Alg. 1) and then I'd expect CCS to have a more-or-less uniform distribution for the rest of the data (i.e. to be a scaled version of "all data") as you claim to take samples from all strata, not just a few. I'm not sure what I'm missing here.
> >
> > > I have a hard time figuring out how the stratified sampling part is different from normal random sampling.
> >
> > > How CCS selects the green part in Fig.4?
> >
> > Thank you for pointing out the confusion in the description of Algorithm 1. We updated Algorithm 1 to include additional explanations and details and added a discussion on the difference between random sampling and stratified sampling to Section 3.3. We summarize the key points below.
> >
> > **The key difference between random sampling and stratified sampling** in Algorithm 1 is as follows. In random sampling, all examples have the same probability to be sampled. In the stratified sampling of Algorithm 1, we divide strata evenly by the difficulty score (line 3 of Algorithm 1). Different strata have the same score range but may have different numbers of examples. Then we assign the same budget to all strata. As a result, examples in a “dense” stratum are likely to have a lower probability of being sampled than the examples in a “sparse” stratum. In random (uniform) sampling, each example would have an equal probability of being sampled.
> >
> > **Algorithm 1 selects the coreset by the following steps:**
> >
> > 1) We first remove hard examples according to the hyperparameter $\beta$ threshold (which leads CCS to have no data on the left side, as shown in Figure 4). This is done prior to stratified sampling.
> >
> > 2)   We then divide the remaining examples into k strata (or bins)  B1,.., Bk, where each stratum covers an equal range of importance scores. For instance, if importance scores were to range from 0 to 100 and k = 10, we would form 10 strata of equal range on scores (e.g., [0,10), [10, 20), …[90,100)) and then assign examples to the strata based on importance score. Such a situation is shown on the CIFAR10 dataset in Figure 4. Each stratum has the same width with respect to the x-axis in Figure 4, but can have a different number of examples
> >
> > 3) We then assign an equal budget to each stratum to achieve the desired pruning rate,  We initialize the coreset to be an empty set. We then do the steps below.
> >
> > 4) We start with the stratum with the fewest examples and randomly select examples from that stratum up to a  minimum of two values: (1) assigned budget or (2) number of examples in that stratum. We add those examples to the coreset. This stratum is removed and the total available budget is reduced by the number of examples sampled.
> >
> > 5) We reallocate the remaining budget equally among the remaining strata.
> >
> > We repeat steps 4 and 5, till no strata remain and return the final coreset.
> >
> > Thus, when a stratum has fewer examples than the assigned budget, all examples will be sampled, which is what occurs in the left part of the green area in Figure 4 (around AUM of 140). Then the remaining strata share the unused budget evenly, resulting in budgets that are slightly larger than on the left part of the green area. Around AUM > 150, the strata have more examples than their assigned budget.  As a result, budgets for the remaining strata do not increase further and thus there is **no peak** on the right side. We see all strata for AUM > 150 end up with the same budget.
> >
> > Graphically speaking, SOTA methods (pruning easy) prune data **from right to left** in Figure 4. CSS first prunes data **from left to right** and then prune data **from top to bottom**. Random sampling gets **a scaled version** of “all data”.
> >
> > We added additional explanations to Algorithm 1 to make this clearer in Section 3.3 of the revised paper. We hope that this response and revisions in Section 3.3 improves the clarity and address your confusion about the algorithm.
> >
> > ---
> >
> > We thank the reviewer again for the constructive review and valuable comments. We appreciate that the reviewer finds the paper to provide valuable novelty. Please let us know if we can provide more information. We hope you can kindly consider updating the rating positively once you feel satisfied with the changes.

---

### Official Review · Reviewer_Vae7 · 2022-10-25

**Confidence:** 4
**Clarity, Quality, Novelty And Reproducibility:** The paper is well-written, and the pr…
**Correctness:** 3
**Technical Novelty And Significance:** 3
**Empirical Novelty And Significance:** 3
**Recommendation:** 5

**Strength And Weaknesses:**

Strength:

1. Selecting subsets to retain the full training performance is important for efficient training, hyperparameter tuning as well as. The problem this paper trying to address is important.

2. The authors do a good job presenting their intuitions developing CCS.

3. The proposed method seems to be technically sound. Theoretical analysis on the risk of training only on subset is provided.

4. The paper is clearly written. Experiments and ablation studies are well performed.  Many details, including the implementation are provided for reproducing.

Weakness:

1. Coverage, or “data representativeness” is widely discussed in the very related active learning literature [1][2]. Active learning shares a lot in common with data selection considering scoring examples and maintaining representative samples. In this field, informativeness (data importance) and representativeness (coverage) are explicitly formulated as two main considerations. This undermines the novelty of this work.

2. There lacks experiments on large scale dataset like ImageNet. The selection strategy behaves pretty inconsistent on ImageNet and Cifar-10, results on small data may not be reliable enough. Besides, ImageNet experiments are commonly conducted in recent data selection methods [3,4,5], so I recommend experiments on ImageNet to further strengthen the paper.

References:

[1] J T. Ash et.al. Deep Batch Active Learning by Diverse, Uncertain Gradient Lower Bounds. ICLR ‘20

[2] G.Citovsky et.al. Batch Active Learning at Scale. NeurIPS ‘21

[3] K. Killamsetty et.al. GRAD-MATCH: Gradient Matching based Data Subset Selection for Efficient Deep Model Training. ICML’21

[4] Ben Sorscher et.al. Beyond neural scaling laws: beating power law scaling via data pruning. arXiv: 2206.14486

[5] Cody Coleman et.al. Selection via Proxy: Efficient Data Selection for Deep Learning. ICLR’19


**Summary Of The Paper:**

This paper tries to ameliorate the serious performance degradation when a substantial portion of data is pruned. It is argued that the problem is caused by poor coverage of the selected subset. Then a new data selection strategy CCS is then proposed. CCS jointly considers sample coverage and the informativeness of each example. The effectiveness of CCS is validated on Cifar-10 and Cifar-100.


**Summary Of The Review:**

The paper is overall well written, however, the problems mentioned in the weakness part undermine its significance.

---

> ### Author Response · Authors · 2022-11-13
> **Response for Reviewer Vae7**
>
> Thank you for your constructive review! We address your questions and concerns below:
>
> > 1. Coverage, or “data representativeness” is widely discussed in the very related active learning literature [1][2]. Active learning shares a lot in common with data selection considering scoring examples and maintaining representative samples. In this field, informativeness (data importance) and representativeness (coverage) are explicitly formulated as two main considerations. This undermines the novelty of this work.
>
> We thank the reviewer for pointing us to connections between our work and active learning work related to data coverage. We have now updated the Related Work (Section 5) discussion to acknowledge that and make the contrast with data coverage in one-shot coreset selection clearer.
>
> While our scheme shares the idea of improving coverage with these methods, one-shot coreset selection philosophically differs from dataset selection in active learning: one-shot coreset selection aims to select a  model-agnostic coreset for future training, but data selection in active learning aims to select the most informative  subset, given the current model state, *for the next iteration of training of that model*. For example, BADGE Ash et al. (ICLR 2020) [1], a diversity-based selection method in active learning, selects a dataset by performing k-means++ on the gradient embedding based on the latest model parameters and then repeats that for subsequent iterations as the model is updated. Similarly, the algorithm in Citovsky et al. (NeurIPS 2021) [2] uses the latest checkpoint to generate embeddings for distance calculation. It is not obvious how either of these coverage strategies is applicable to the one-shot coreset selection.
>
> The above philosophical difference has significant implications. In active learning,  a batch of examples that are informative at one iteration of a model may not be informative for training another model or even for training another iteration of the same model.  Also, there is no guarantee that a batch or set of batches from active learning will serve as a good one-shot coreset for training new models from scratch.
>
> Our proposed method based on stratified sampling across importance scores is also novel for one-shot coreset selection. To the best of our knowledge, stratified sampling on importance scores has not been previously considered for the coreset selection problem. As acknowledged by all the reviewers, CCS significantly outperforms not only SOTA methods but also random sampling at high pruning rates.
>
> Thus, we hope we have just made a persuasive case as to why our study on catastrophic accuracy drop and the CCS algorithm is a worthy contribution to the one-shot coreset selection field.
>
>
> > 2. There lacks experiments on large scale dataset like ImageNet. The selection strategy behaves pretty inconsistent on ImageNet and Cifar-10, results on small data may not be reliable enough. Besides, ImageNet experiments are commonly conducted in recent data selection methods [3,4,5], so I recommend experiments on ImageNet to further strengthen the paper.
>
> We agree with the reviewer that evaluation on ImageNet can better support our claim and make our argument more convincing. We now added the evaluation results on **ImageNet**. Here are the results for AUM metric on ImageNet:
>
> | Pruning rate | 30%   | 50%   | 70%   | 80%   | 90%   |
> |--------------|-------|-------|-------|-------|-------|
> | Random       | 72.18 | 70.34 | 66.67 | 62.54 | 52.34 |
> | AUM          | 72.53 | 66.57 | 40.42 | 21.12 | 9.93  |
> | AUM (CCS)    | 72.29 | 70.52 | 67.78 | 64.47 | 57.36 |
>
> The evaluation results show that SOTA methods still experience the catastrophic accuracy drop problem at high pruning rates, and CCS outperforms SOTA methods as well as random sampling at high pruning rates. The finding on ImageNet is consistent with findings on CIFAR10, CIFAR100, SVHN, and CINIC10.  (SVHN and CINIC10 results were in the Appendix due to space constraints.)
>
> We have added ImageNet evaluation results to Appendix D.4.
>
> ---
>
> We thank the reviewer again for pointing us to work in active learning, which we now include in the paper (Section 5), and suggesting the ImageNet experiments, which also have been added to the paper (Appendix D.4). We hope our response addressed your concerns about the novelty and contribution. Please let us know if we can provide more information related to your concerns. We hope you can kindly consider updating the rating positively once you feel satisfied with the changes.
>
> -----
> [1] J T. Ash et.al. Deep Batch Active Learning by Diverse, Uncertain Gradient Lower Bounds. ICLR ‘20.
>
> [2] G.Citovsky et.al. Batch Active Learning at Scale. NeurIPS ‘21.

---

> ### Author Response · Authors · 2022-11-30
> **Thanks to Reviewer Vae7 and Response Summary**
>
> Dear Reviewer Vae7,
>
> We greatly appreciate your time in reviewing our paper again. We would like to kindly remind you that the response and the revised paper are ready.
>
> *One of the reviewers replied that our responses successfully addressed the concerns and confusion and believes the paper to be above the threshold for acceptance.* We hope our responses can also address your concerns. We are looking forward to your feedback!
>
> **We would like to briefly summarize the response to your major concerns for your convenience.**
>
> > Coverage is already discussed in the area of active learning.
>
> In our previous response, we agreed with the reviewer that data coverage is not a new thing in active learning, but we showed that this factor is overlooked in the area of one-shot coreset selection. We also added more discussion to Related Work on active learning to discuss this point.  A key point is data selection in active learning and one-shot coreset selection are very different problems.  Furthermore, besides the baselines discussed in our paper, *our method also outperforms another submission to ICLR2023*, which receives quite positive ratings (https://openreview.net/forum?id=7D5EECbOaf9), which implies that our method design is not trivial or obvious.
>
> > Lack of evaluation on ImageNet:
>
> We added the evaluation results on ImageNet, and we found that our method still outperformed the baselines on ImageNet. Thank you for suggesting that.
>
> ---
>
> We have already revised the paper to include these points in the paper. We also kindly refer you to our previous responses (both separate and general responses) for more detailed discussions and responses to your other concerns. Please let us know if we can provide more information to better address your concerns and clarify the claim in the paper!

---

### Official Review · Reviewer_f2Gh · 2022-11-04

**Confidence:** 4
**Correctness:** 3
**Technical Novelty And Significance:** 2
**Empirical Novelty And Significance:** Not applicable
**Recommendation:** 5

**Clarity, Quality, Novelty And Reproducibility:**

The paper is well written and its original as far as application of stratified sampling to the pruning of training examples in combination with importance score is concerned. However, the stratified sampling per-se is a common approach in this domain.

**Strength And Weaknesses:**

Strength: A well defined practical problem is solved using a simple method. An investigation of the SOTA methods revealed the problem with the existing approaches in high pruning regime, and a simple stratified sampling based fix solves the problem. Though the theory is not directly relevant to the proposed approach and is not innovative, but the extension of r-cover core-set bounds to the p-partial r-cover bounds does show the tradeoff between p-partial cover and r-radius cover. The numerical experiments show a significant improvement over the baseline methods in the high pruning regime.

Weakness: The paper lacks novelty, and the given theory is very tangential to the proposed solution. A theoretical bound on loss function for stratified sampling in combination with any of the SOTA importance score estimation method would be more useful.

**Summary Of The Paper:**

This paper addresses the following problem. Given a set of training examples, select a smaller subset of it which minimizes the expected model error. The goal is that the smaller subset can be used to train a different model with minimal drop in accuracy in comparison to training the model using the entire training dataset.

The SOTA approaches solve the problem in the following way: They compute an importance score for each of the training example for the given model. The examples with smaller importance score are pruned as they are not important enough for learning of the model. The importance score is computed in different ways. 1/ Forgetting score (Toneva et al., 2018) is defined as the number of times an example is incorrectly classified after having been correctly classified earlier during model training. 2/ Area under the margin (AUM) (Pleiss et al., 2020): AUM represents the probability gap between the target class and the largest other class across all training epochs. A larger AUM means higher difficulty and importance. 3/  EL2N (Paul et al., 2021): EL2N scores estimate data difficulty by the L2 norm of error vectors. Examples with large difficulty score are more important. 4/ Entropy (Coleman et al.,2019): The entropy of outputs reflects the uncertainty of training examples, and a high entropy indicates an example containing more information and is more important.

The above described methods beats the baseline random selection method in moderate pruning regime but performs poorer than random method in high pruning regime, say 90% pruning.

The paper identifies that poor performance of the SOTA methods in comparison to random selection is because SOTA methods prune all the easy examples which correspond to the high density region of training examples. Hence the paper proposes stratified sampling instead of pruning all the low-score examples. It implemented stratified sampling on top of each of the SOTA methods described above and shows that it significantly improves performance in high pruning regime in comparison to the low-score pruning of the SOTA method.

The paper also extends the Lipschitz constant based bounds on pruning based loss function when pruning is done using core-set radius r, to the p-partial r-cover based bounds, where radius r covers only p fraction of the probability density space.


**Summary Of The Review:**

The paper provides a simple approach for a well defined problem in high pruning regime. However, the approach lacks novelty and the provided theory is very tangential to the proposed approach.

---

> ### Author Response · Authors · 2022-11-13
> **Response for Reviewer f2Gh**
>
> Thank you for your constructive review! We address your questions and concerns below:
>
> >The given theory is very tangential to the proposed solution. A theoretical bound on loss function for stratified sampling in combination with any of the SOTA importance score estimation methods would be more useful.
>
> The theory is designed to give us insights into the relationship between coverage and accuracy drop. We agree that the theorem does not directly translate to the algorithm, but it does help provide a crucial direction for our proposed method and got us focused on improving coverage. The stratified sampling idea came to us when we were thinking about how best to get a high $AUC_{pr}$ score over a coreset. As we point out in Section 4 and in more detail in Appendix D.2, the proposed CCS method also improves $AUC_{pr}$ given the same pruning rate.
>
> The theoretical analysis in our paper also allows us to show that SOTA methods generally overlook data coverage based on this metric at high pruning rates, and the proposed algorithm addresses that gap.  Besides, we also note that in the theoretical analysis, we extend the geometric set cover problem to a density-based distribution cover setting. The p-r tradeoff curve and the $AUC_{pr}$ metric are both novel ways to view coverage and provide a valuable concrete viewpoint to inspire future research for the community.
>
> We agree that it will be useful to find a bound on loss function for stratified sampling on importance scores. We considered that, but it is a challenging problem because it is difficult to characterize the mathematical properties of commonly-used heuristic importance metrics. This would be a fascinating research problem to investigate more deeply in the future.
>
> >The paper lacks novelty.
>
> > its original as far as application of stratified sampling to the pruning of training examples in combination with importance score is concerned. However, the stratified sampling per-se is a common approach in this domain.
>
> We agree with the reviewer that stratified sampling is a well-known technique in the sampling domain in general. Stratified sampling by itself is not our claim to novelty.
>
> What is novel about our work is recognizing a relationship between coverage and catastrophic accuracy drop that is observed with SOTA techniques for one-shot coreset selection for deep learning. The one-shot coreset selection problem for deep learning has been studied for many years (see Reference [3] below from 2018), but the catastrophic accuracy drop issue has not been addressed. To the best of our knowledge, our paper is the first to conduct an in-depth study on this problem and propose a novel method that addresses this issue.
>
> It is also novel that we were actually able to address the catastrophic accuracy drop by leveraging a simple idea of stratified sampling. The conventional wisdom in the coreset community is to bias selection strongly towards high-importance examples. Our work presents a different perspective to the research community that, instead,  diversity across importance scores is also very important.
>
> The CCS performance results, as mentioned by all reviewers, are significant. Beating random at high pruning rates   is non-trivial and has been elusive in past work on one-shot coreset selection and CCS achieves that. CCS also performs comparably to existing excellent baseline methods at low pruning rates. This is a significant contribution to the field and provides a new baseline for coreset selection research. We have released its source code publicly in anonymized form so that the community can make use of it.
>
> We kindly refer to the general response for additional discussions on this issue.  We have updated the contents of the paper to hopefully bring out the novelty aspects more clearly in Section 1 and Section 5.
>
> ----
>
> Thank you again for the constructive review. We hope our additional explanation addresses your concerns about the novelty and contribution. Please let us know if we can provide more information related to your concerns. We hope you can kindly consider updating the rating positively once you feel satisfied with the changes.
>
> ----
>
> [3] Toneva, Mariya, et.al.  "An Empirical Study of Example Forgetting during Deep Neural Network Learning." ICLR ‘18

---

> ### Author Response · Authors · 2022-11-30
> **Thanks to Reviewer f2Gh and Response Summary**
>
> Dear Reviewer f2Gh,
>
> We greatly appreciate your time in reviewing our paper again. We would like to kindly remind you that the response and the revised paper are ready.
>
> *One of the reviewers replied that our responses successfully addressed the concerns and confusion and believes the paper to be above the threshold for acceptance.* We hope our responses can also address your concerns. We are looking forward to your feedback!
>
> **We would like to briefly summarize the response to your major concerns for your convenience.**
>
> > The given theory does not directly contribute to the proposed method.
>
> In our previous response, we explained that the goal of theory is to test the hypothesis that bad coverage is an essential reason for accuracy drop, which provides crucial motivation and intuition for designing our method.
>
> > Stratified sampling is commonly used in this area.
>
> We explained in our previous response that, as far as we know,  we are the first to propose such a method in the area of one-shot coreset selection for deep learning. Besides, our method is a novel design for this specific question and is inspired by the idea behind stratified sampling rather than a direct application of stratified sampling.
>
> ---
>
> We have already revised the paper to include these points in the paper. We also kindly refer you to our previous responses (both separate and general responses) for more detailed discussions and responses to your other concerns. Please let us know if we can provide more information to better address your concerns and clarify the claim in the paper!

---

### Author Response · Authors · 2022-11-13
**Response to all reviewers (Part 1)**

Dear reviewers,

We thank all the reviewers for their constructive reviews and are very grateful for their feedback towards improving our work. We refer Reviewer f2Gh as Reviewer #1, Reviewer Vae7 as Reviewer #2,  Reviewer 17rD as Reviewer #3, and Reviewer 6QHN as Reviewer #4 for simplicity. (It is the order of the reviewers on the website.)

We are pleased that reviewers found our paper’s advantages: “well-defined and important problem”(Reviewers #1, #2), “Well-grounded and technical sound methods”(Reviewers #2, #4), “significant improvement over the baseline methods” (All reviewers), “novel theoretical analysis contribution” (Reviewers #3), “well-written” (All reviewers).

For the comments and concerns discussed in the reviews, we write this general response and separate responses to each individual review. Besides, we have revised the manuscript to improve the presentation of the paper and address the reviewer’s concerns as well.

---------

The major questions from reviewers are on:

1. novelty on the basis of data coverage being also discussed in the active learning scenario (Reviewer #2) and stratified sampling being used in other contexts (Reviewer #1);
2. possible existence of a more natural and direct solution such as the sampling in proportion to importance  (Reviewer #4);
3. lack of evaluation on ImageNet (Reviewer #2 #3).

We address each below.

### 1. Novelty

**1) Novelty with respect to active learning:**

We thank Reviewer #2 for pointing us to work on data coverage in active learning. We have added an acknowledgment to that effect and a comparison with data selection in active learning to Related Work.

While our scheme shares the idea of improving coverage with these methods [1][2], one-shot coreset selection philosophically differs from dataset selection in active learning, leading to significant differences in how data selection is done. One-shot coreset selection aims to select a *model-agnostic coreset* for training new models from scratch, while active learning aims to select a subset that, given the current state of the model, improves the model in the next training round. For example, BADGE Ash et.al. [1], a diversity-based selection method in active learning, selects a subset by performing k-means++ on the gradient embedding based on the latest model state and then repeats the selection process for subsequent rounds during training.  It is not clear how to apply this idea for getting coverage in active learning to the one-shot core selection problem.

Another significant difference is that in active learning, the total number of examples queried across all rounds of training can be much larger than the size of the desired one-shot coreset. For instance, in the experimental results of Ash et.al. [1], BADGE queried a different set of 10K (20%) examples on every iteration of training for the CIFAR-10 dataset. The total number of different examples queried across all iterations is even larger.  In contrast, CCS finds a fixed small subset (e.g. only 10% of the dataset at a 90% pruning rate) that can be used to train new models from scratch while achieving high accuracy.

To clarify our contributions more clearly, we have now added a discussion on active learning in Section 5 (related work). We thank the reviewers for pointing us to the related coverage work in the active learning community.



**2) Novelty with respect to stratified sampling:**

We agree with Reviewer #1 that stratified sampling is a well-known technique in statistics. That is not our claim to novelty.

What is novel about our work is recognizing a relationship between coverage and catastrophic accuracy drop that is observed with SOTA techniques for one-shot coreset selection for deep learning. The one-shot coreset selection problem for deep learning has been studied for many  years (e.g., see work [3]), but the catastrophic accuracy drop issue has not been addressed. To the best of our knowledge, our paper is the first to conduct an in-depth study on this problem and propose a novel method that addresses this issue.

It is also novel that we were actually able to address the catastrophic accuracy drop by leveraging a simple idea of stratified sampling. The use of simpler and well-established techniques from other or related fields to solve a challenging problem should be considered a strength, not a weakness.  That required understanding the problem from first principles to develop the insight. The conventional wisdom in the coreset community is to bias selection strongly towards high-importance examples and our work presents a different perspective.

Also note that we needed to customize stratified sampling for the coreset selection problem.  For instance, we first pruned some hardest examples (according to hyperparameter $\beta$) based on the properties of coresets. As far as we are aware, we are the first to propose stratified sampling over importance scores for one-shot coreset selection.

---

> ### Author Response · Authors · 2022-11-13
> **Response to all reviewers (Part 2)**
>
> ### 2. Can proportional importance sampling be a more natural and direct solution?
>
> We thank Reviewer #4 for suggesting proportional importance sampling and agree that such sampling should also improve coverage over SOTA methods. We decided to evaluate proportional importance sampling on CIFAR10. We mapped importance scores to probabilities using a softmax function and then picked examples based on their probabilities to form the coreset.
>
> Unfortunately, the proportional importance sampling method underperforms random sampling at high pruning rates; see the AUM (Importance Sampling) row in the table below.  It does better than a SOTA method (AUM), but not random sampling. Beating random selection is crucial for any coreset selection algorithm since random selection is the simplest selection strategy. We add these evaluation details on the proportional importance sampling in Appendix E.2 and mention it in the Evaluation section.
>
>
> | Pruning rate              | 30%   | 50%   | 70%   | 80%   | 90%   |
> |---------------------------|-------|-------|-------|-------|-------|
> | Random                    | 94.33 | 93.40 | 90.94 | 87.98 | 79.04 |
> | AUM                       | 95.07 | 95.26 | 91.36 | 57.84 | 28.06 |
> | AUM (CCS)         | 95.27 | 94.93 | 93.00 | 90.91 | 86.08 |
> | AUM (Importance Sampling) | 93.12 | 92.61 | 90.77 | 86.27 | 77.60  |
>
> The above experiment results suggest that it is non-trivial to beat random sampling at high pruning rates, even with a scheme that improves data coverage. Earlier in our work, we ourselves wondered whether any scheme could beat random at high pruning rates since all existing SOTA methods were underperforming random selection at high pruning rates.
>
> CCS achieves that remarkable result  -- it beats random consistently for five datasets at high pruning rates and also beats SOTA methods by an even larger margin.  We therefore believe that CCS is a valuable contribution to the research community.
>
> We do not rule out that there could be a better way to map importance scores to probabilities and to make proportional importance sampling, as suggested by Reviewer #4,  perform better.  But, CCS uses stratified sampling and performs really well.  CCS thus provides a good baseline for future work in this direction.
>
> ### 3. Evaluation on ImageNet
>
> We agree with Reviewer #2 and Reviewer #3 who suggest that evaluating CCS on **ImageNet** can better support our claims. We present the evaluation results for AUM metric on ImageNet below:
>
> | Pruning rate | 30%   | 50%   | 70%   | 80%   | 90%   |
> |--------------|-------|-------|-------|-------|-------|
> | Random       | 72.18 | 70.34 | 66.67 | 62.54 | 52.34 |
> | AUM          | 72.53 | 66.57 | 40.42 | 21.12 | 9.93  |
> | AUM (CCS)    | 72.29 | 70.52 | 67.78 | 64.47 | 57.36 |
>
> The evaluation results show that SOTA methods still experience the catastrophic accuracy drop problem at high pruning rates, and CCS outperforms SOTA methods as well as random sampling at high pruning rates.
>
> We have added ImageNet evaluation results to Appendix D.4.
>
> Finally, we provide additional details in individual responses to each of the reviewers’ comments.
>
> ---
>
> ### Revision Summary
>
> We also revised the paper based on reviewers’ comments and polished the writing even further where possible.  Significant changes that address reviewer comments specifically are highlighted in blue. We also cited all references mentioned by the reviewers in the revised paper.
>
> Here is the summary of the changes that address reviewer comments in the revised paper:
>
> Section 1 (Introduction): Updated the introduction to better discuss our contribution and novelty in the one-shot coreset selection problem for deep learning.
>
> Section 2 (Preliminaries): Updated Equation.1 to formulate it in terms of pruning rate $\alpha$.
>
> Section 3.3 (Methodology): Update the algorithm description for better clarity.
>
> Section 4 (Evaluation): Added additional explanation on baselines and CCS.
>
> Section 4.2 (Ablations):  Clarified the sensitivity to the hyperparameter $k$.
>
> Section 5 (Related work): Updated the related work to address novelty and difference from work on active learning and coreset selection in clustering problems.
>
> Appendix B (Experiment setting): Added setting for the experiments on ImageNet.
>
> Appendix D.4 (Evaluation): Added evaluation results on ImageNet.
>
> Appendix D.5 (Evaluation): Added results and discussion on the proportional importance sampling method.
>
> ----
> We thank all the reviewers for their constructive feedback hat has helped us improve the paper. We hope we addressed their concerns and kindly request them to update their review and scores accordingly. We welcome additional comments.
>
> ----
> [1] J T. Ash et.al. Deep Batch Active Learning by Diverse, Uncertain Gradient Lower Bounds. ICLR ‘20.
>
> [2] G.Citovsky et.al. Batch Active Learning at Scale. NeurIPS ‘21.
>
> [3] Toneva, Mariya, et.al.  "An Empirical Study of Example Forgetting during Deep Neural Network Learning." ICLR ‘18

---

### Author Response · Authors · 2022-11-17
**Response follow-up**

Dear all reviewers,

We greatly appreciate your comments on our paper again, and we would like to kindly remind you that we have uploaded the response and the revised paper. We hope that our responses answer your questions and address your concerns.

Please let us know if we can provide more information. Looking forward to your feedback.

---

### Author Response · Authors · 2022-11-26
**Response follow-up and new comparison to a concurrent submission in ICLR2023**

Dear all reviewers and ACs,

We greatly appreciate your time in reviewing our paper again. Since it is quite close to the end of the phase 2 discussion, we would like to kindly remind you that the response and the revised paper are ready. We are looking forward to your feedback to discuss your concerns on the paper.

We would like to note that **our paper addresses a well-defined, important, and actively-studied problem in the one-shot coreset selection for deep learning.** We strictly follow other SOTA papers to make the baseline comparison and show remarkable performance improvement.

We thank the reviewers for pointing out areas where they felt discussion of related work could be better to address their concerns above novelty. We think the issue is totally fixable, and we have already fixed that in the posted revision. We also improved the paper based on the reviewers’ suggestions.

### Comparison to another concurrent work on the same problem submitted in ICLR2023:

As a side note, besides SOTA schemes discussed in our paper, we found that our scheme also remarkably outperforms at all pruning rates even the scheme proposed in **a concurrent submission (Moderate) in ICLR2023 [1] on one-shot coreset selection for deep learning**, which gets generally positive ratings (https://openreview.net/forum?id=7D5EECbOaf9).

We evaluated both on the CIFAR100 dataset, since that was a common dataset in both papers. Since the detailed training settings are different in the two papers, we report both accuracies reported in their paper and with our training setting in the below table. The accuracy comparison shows that our proposed method also **remarkably outperforms** the method proposed in this concurrent submission:

| Pruning rate | 30%   | 50%   | 70%   | 80%   | 90%   |
|--------------|-------|-------|-------|-------|-------|
| Random       | 74.59 | 71.07 | 65.30 | 57.36 | 44.76 |
|Moderate in reported [1]        | ~73.00 | ~68.00 | ~57.50 | ~52.00 | N/A  |
|Moderate with our training setting    | 74.64 | 71.99 | 65.11 | 59.11 | 48.66 |
|Our method (AUM)    | 76.84 | 73.77 | 68.85 | 63.20 | 55.03 |

The above suggests that one-shot coreset selection is a problem of active research interest and our scheme sets a new baseline for one-shot coreset selection at high pruning rates, even considering schemes that are concurrently submitted. Thus, we believe that our paper provides a worthy contribution to the research community.

**We will really appreciate the reviewers’ help if reviewers could spend a bit extra time to let us know what other concerns they may have and give us another opportunity to improve our paper further.**

[1] Anonymous authors. “Moderate Coreset: A Universal Method of Data Selection for Real-world Data-efficient Deep Learning” Submitted to ICLR 2023 https://openreview.net/forum?id=7D5EECbOaf9

---

### Decision · Program_Chairs · 2023-01-20

**Decision:**

Accept: poster

**Justification For Why Not Higher Score:**

Authors provided extensive clarifications and resolved the concerns raised by the reviewers. As indicated by one of the reviewers this work relates to work in the area of active learning and should better articulate the novelty. One of the reviewers who voted for acceptance acknowledged that they had initially not fully understood the proposed method.

**Justification For Why Not Lower Score:**

The problem the authors consider is an important one and the proposed method is worth sharing with the community. The experimental evaluation supports the claims made by the authors and the reported results are strong.

**Metareview: Summary, Strengths And Weaknesses:**

This work considers coreset selection in the context of high pruning regime. The drop in accuracy in this regime is well known and an unresolved problem. The authors show that importance-based selection scores are detrimental. The proposed method based on stratified sampling increases data coverage, which is shown to be a determining factor in practice. The empirical results reported by the authors are convincing and the additional experiments conducted during the rebuttal (including a comparison to a work on the same problem submitted in ICLR 2023) further added evidence of the strong performance.

**Note From Pc:**

if the above contains the word "oral" or "spotlight" please see: "oral" presentation means -> notable-top-5% and "spotlight" means -> notable-top-25%. As stated in our emails, we are disassociating presentation type from AC recommendations

**Summary Of Ac-Reviewer Meeting:**

N/A